# Evolution and transmission dynamics of wild poliovirus in Pakistan and Afghanistan (2012-2023)

David Jorgensen [1] ✉, Margarita Pons-Salort [1], Muhammad Salman[2], Adnan Khurshid[2], Yasir Arshad[2], Nayab Mahmood[2], Darlan da Silva Candido[1], Steve Kroiss[3], Hil Lyons[3], Nicholas C. Grassly [1] & Muhammad Masroor Alam[2]

Despite concerted global vaccination efforts, wild poliovirus remains endemic in two countries in 2024, Pakistan and Afghanistan. This study uses phylo-geographic analysis of poliovirus genetic and epidemiological data from clinical and wastewater surveillance to identify the causes of poliovirus per-sistence and routes of spread over the last decade (2012 to 2023). Poliovirus genetic diversity declined after 2020, with one of two major genetic clusters dying out, and recent detections are now closely related genetically. High-risk and hard-to-access regions have sustained polio transmission over the past decade, even when interrupted elsewhere. Karachi, one of the most densely populated cities globally, has acted as a hub for the amplification and spread of poliovirus to other regions, many of which we show to be dead-end for onwards transmission despite frequent virus detection. Phylogenetic analysis has long been central to the polio surveillance network, and advancing the approaches used can provide critical epidemiological insights to accelerate eradication efforts.

Since the launch of the Global Polio Eradication Initiative (GPEI) in 1988, the world has come close to eliminating wild poliovirus (WPV), with cases plummeting from hundreds of thousands annually to just a handful today. Two of the three WPV serotypes have been eradicated, and only serotype 1 (WPV1) remains in circulation, endemic to just two countries: Pakistan and Afghanistan.

Despite significant progress, these countries have never completely interrupted WPV1 transmission. The longest period without any reported cases began in early 2021, with Afghanistan going eight months and Pakistan fourteen months without new poliomyelitis cases[1,2]. During these periods, poliovirus continued to be detected in wastewater, indicating that the virus was still silently circulating in under-immunized populations, even in the absence of clinical cases. Reaching the goal of a polio-free world hinges on stopping WPV1 transmission in its final strongholds.

Between 2012 and 2023, there were two major outbreaks of WPV1 in Pakistan and Afghanistan, around 2013–14 and 2019–2020 (Fig. 1A). As of the end of September, there have been 58 reported paralytic cases in 2024 across the two countries, a significant expansion on the 12 reported cases in 2023 and the highest number in a single year since 2019. This highlights the continued threat of WPV1 transmission for the health of children in the region and globally. As recently as 2022, Mozambique and Malawi reported a 9-case WPV1 outbreak, linked to transmission in Southern Pakistan[3]. The persistence of wild poliovirus and associated risk of international spread led polio to be designated a Public Health Emergency of International Concern (PHEIC) by the World health Organization in 2014[4].

Understanding WPV1 circulation in Pakistan and Afghanistan is complex due to several interacting factors which both help to support viral circulation and make eradication efforts difficult. Insecurity in the

[1]MRC Centre for Global Infectious Disease Analysis, Department of Infectious Disease Epidemiology, School of Public Health, Imperial College London, London, UK. [2]National Institute of Health, Islamabad, Pakistan. [3]Institute for Disease Modeling, Global Health Division, Gates Foundation, Seattle, WA, USA. ✉e-mail: david.jorgensen13@imperial.ac.uk

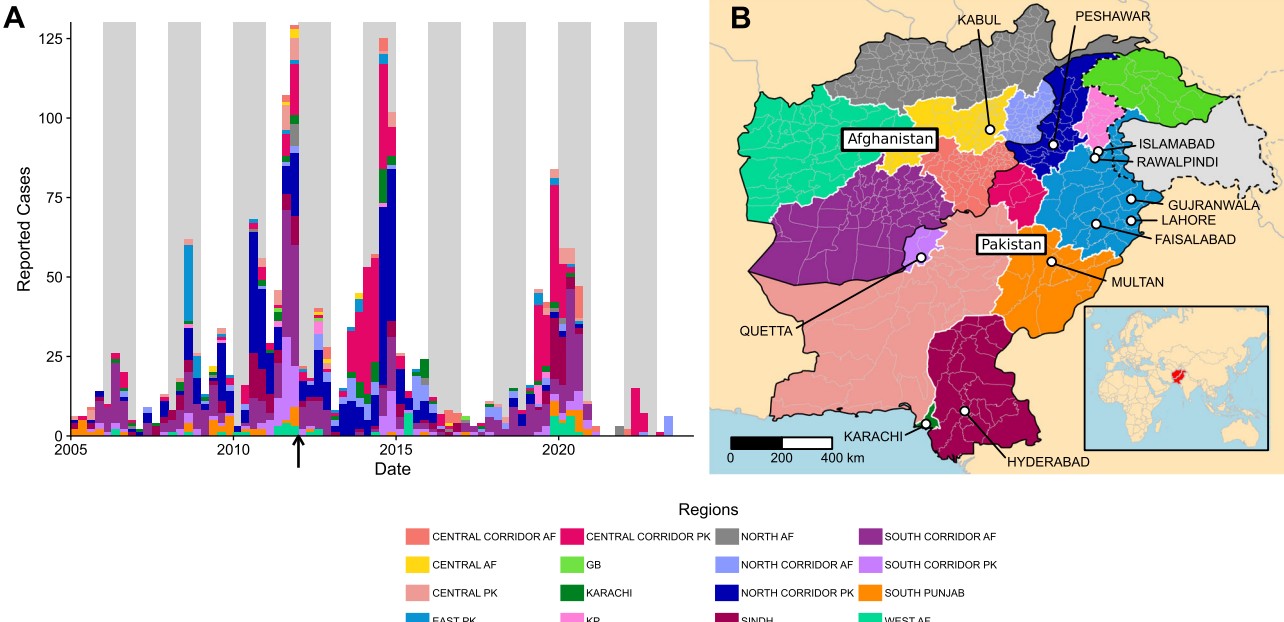

**Fig. 1 | Reported cases of acute flaccid paralysis (AFP). A** Reported AFP cases in Afghanistan and Pakistan aggregated in 3-month periods between January 2005 and August 2023. Gray strips show even years. The black arrow on panel A shows the sampling time of the first genetic sequence included in this analysis. **B** Map showing the 16 regions used for the phylogeographic analysis, with cities of over 10 million inhabitants plotted as white points and labeled. Disputed territory is shown with dotted lines. The geodata used is obtained from the World Health Organization. The boundaries and names shown, and the designations used on the map do not imply the expression of any opinion whatsoever on the part of the authors or the World Health Organization concerning the legal status of any country, territory, city or area or of its authorities, or concerning the delimitation of its frontiers or boundaries.

region, including armed conflict, intermittent bans on house-to-house vaccinations, and targeted attacks on vaccinators, have made certain regions and communities difficult or impossible to access for health workers, allowing polio transmission to persist despite overall increases in vaccination coverage[5,6]. Additionally, the porous border between the two countries and movement of populations in the geographically isolated border regions (both nomadic and displaced populations) complicate efforts to contain the spread of WPV1, with some regions hard to reach for both vaccinators and healthcare services[7]. Societal mistrust and misinformation around polio vaccination have also driven vaccine refusals, although overall reported coverage of vaccination campaigns remains high[8,9]. Alongside issues in reaching rural communities, sanitation issues in dense urban populations as well as widespread flooding and monsoon weather, exacerbate seasonal variation in polio transmission.

Understanding patterns of viral spread and persistence is further hampered by a vast majority of poliovirus infections being asymptomatic and by some areas having poor surveillance for paralytic cases. Environmental surveillance (ES) based on testing wastewater samples for the presence of poliovirus has been widely implemented in the region to help detect circulation of poliovirus in the absence of cases. ES data have been shown to provide early warning of circulation approximately 4 months prior to surveillance using acute flaccid paralysis (AFP) (a rare and severe outcome of poliovirus infection) alone in Pakistan[10]. In order to target vaccination responses to endemic circulation of WPV1 it is important to understand past transmission patterns in the region, identifying which areas support long-term persistence of viral circulation and act as sources for infection elsewhere. Here, we analyze routinely collected poliovirus genetic sequences from the VP1 region of the viral capsid collected over more than a decade (2012 – 2023) from both patients expressing symptoms of AFP and ES.

As the VP1 region is sequenced routinely for genotyping and serotyping of detected viruses, a wealth of poliovirus VP1 sequences are produced globally[10–17]. So far, WPV1 sequences have primarily been

used in the GPEI to routinely identify "genetic clusters" based on pairwise genetic relatedness of isolates[13]. Geographic analysis of the movement of these clusters has been published previously for short time periods, but an overall historic look and inference of virus movement over time and space is missing from the published literature[13]. Here we use a phylogeographic approach to investigate poliovirus spread in Pakistan and Afghanistan, identifying key transmission routes and describing trends in viral diversity and long term persistence over time[18,19].

## Results

### Included genetic sequence data and association with reported cases

Here, we analyze 3843 VP1 sequences from wild-type polioviruses detected in samples collected between January 2012 and June 2023. During this period, there were a far greater number of genetic sequences from ES detections than AFP cases (2727 ES to 1116 AFP) (Supplementary Table S1), and the number of ES sites expanded significantly through the time-period covered (Supplementary Figs. S1 and S2). Throughout this analysis, we divide Afghanistan and Pakistan into 16 regions, defined based on previous epidemiological analysis of polio, other diseases and climatic regions (see map in Fig. 1B and Methods). We provide alternative estimates of key reported parameters obtained with the full dataset (including AFP and ES sequences; presented in the main figures) and using only AFP data, considered a priori less biased than ES detections, which tend to be targeted to large sewage catchments and key high-risk areas for polio transmission[20,21]. Figures for the analysis of the AFP data alone are presented in the Supplementary Information. Correlation analysis with Kendall's Tau revealed moderate positive correlation between the number of sequences and the number of reported WPV1 AFP cases ($r = 0.46$, $p < 0.001$). Further analysis by region (correlation ranges from $r = 1.00$ to $r = -0.11$) and year (ranges from 1 to $-0.33$) suggests the sequence data reporting bias varies widely across regions and years (Supplementary Table S2). This is likely driven by differences in

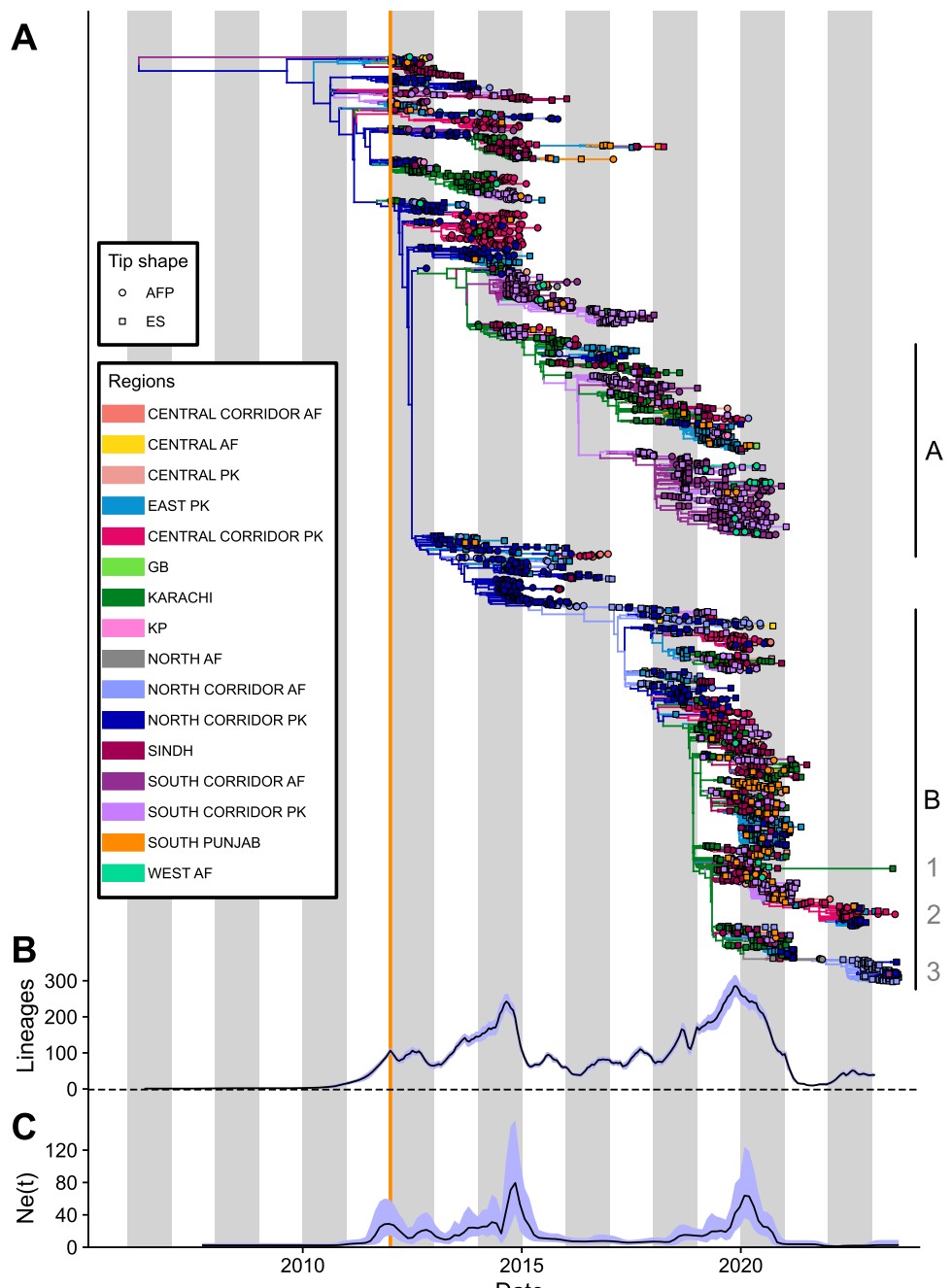

**Fig. 2 | Phylogenetic analysis. A** Dated consensus phylogenetic tree reconstructed from WPV1 sequences from stool of AFP cases and ES samples collected between January 2012 and August 2023. The vertical orange line indicates the time of the first sequence in the database. Branches and internal nodes are colored by their inferred location, and tips, by their known location. The two lineages of poliovirus in the region, which were responsible for the 2019–2020 outbreak, are labelled A and B, with the numbered clusters inside of lineage B supporting all transmission seen in 2023. **B** Lineages through time (LTT) plot inferred from all available sequences. **C** Effective viral population size (Ne) inferred from genetic sequences generated from stool samples from AFP cases. Blue shaded regions on panels B and C show 95% CrIs around the mean estimate shown in black. Vertical gray stripes represent even years.

ES sampling intensity (which affects the number of reported WPV sequences) and vaccination coverage (which affects the number of reported AFP cases) across regions and years. Alongside reporting of key parameters using only AFP sequence data, we additionally present tip-state-swap phylogeographic analysis with the full dataset to quantify the bias introduced by differences in sequencing effort by region.

**Genetic diversity through time**

A phylogeny of all sequences shows two clear lineages (here labeled A and B), which drove all WPV1 transmission during the most recent

outbreak, in 2019–2020 (Fig. 2). The root of these two poliovirus lineages was found to be in mid-2009 (2009-08-22; 95% HPD: 2009-02-27 to 2010-02-02) when including all sequences, with no significant difference when using AFP data alone (2009-04-16; 95% HPD: 2008-02-23 to 2010-01-21). Lineage A died out in early 2021 and has not been detected since. Three distinct sub-lineages of the B lineage persisted into 2023 (Fig. 2).

Bayesian skyline analysis of AFP data inferred increases in viral diversity during the two major outbreaks, followed by a reduction between the outbreaks and a significant decline since early 2021

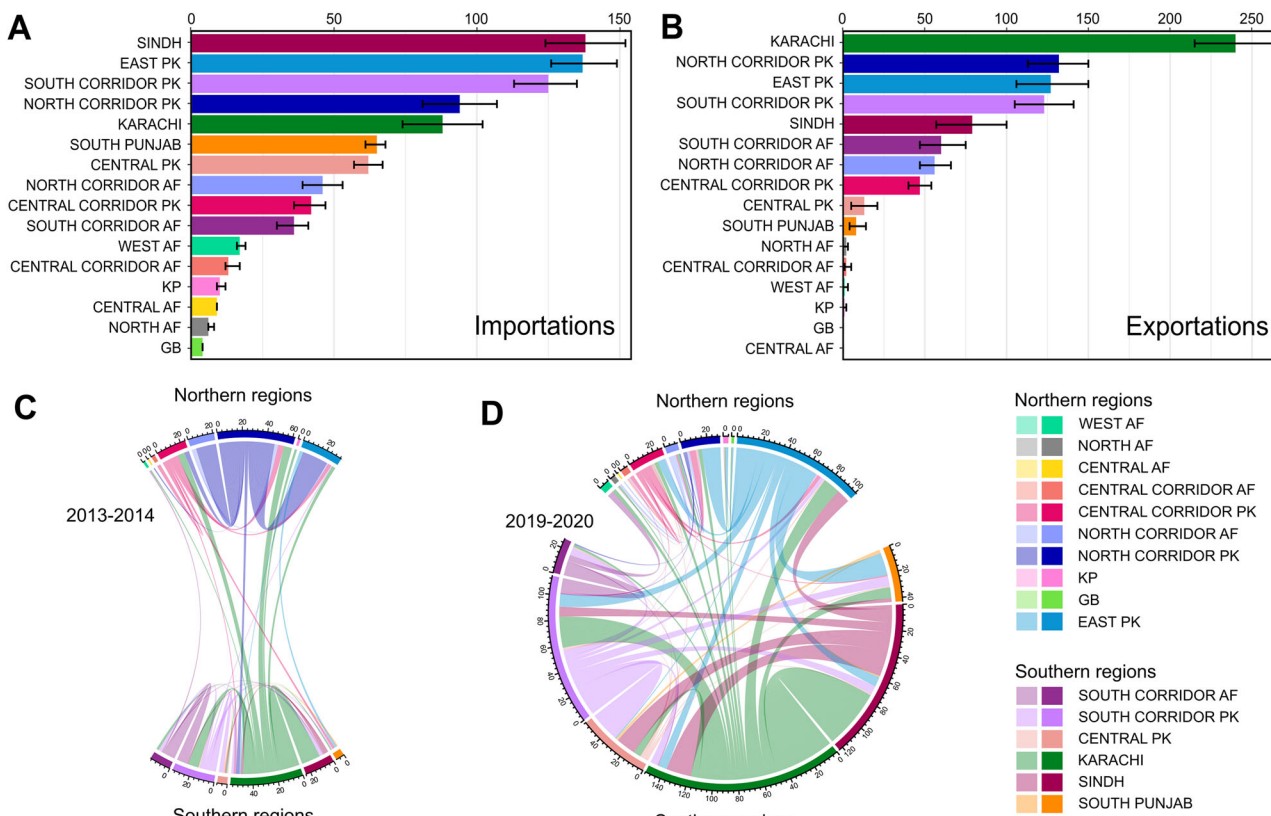

**Fig. 3 | Estimated viral movements across regions. A, B** Bayes factor supported estimates of the number of movements between the regions across the full time period. Panel **A** shows the number of importations into each region and **B** the number of exportations from each region. 95% CrIs for each estimate are shown with black error bars. The values used to generate these bars are shown in Supplementary Table S2. Inferred movement events between regions during the **C** 2013–2014 and **D** 2019–2020 outbreak periods based on the reconstruction of most likely ancestral regions. Links between regions are colored according to the exporting region. **C** and **D** are scaled to show the relative number of movement events between regions in each outbreak period, with regions in the north of each country plotted at the top and regions in the south plotted at the bottom. A matrix representation of the number of movements between regions is provided in Supplementary Fig. S9. Colors are mapped to regions in Fig. 1B.

(Fig. 2C). In contrast, a lineages-through-time (LTT) analysis of both AFP and ES data demonstrated the early-warning capabilities of ES data, detecting an increase in viral lineages ahead of both major outbreaks, even when AFP data showed no such signal. Towards the end of the study period, the LTT analysis indicated expanding transmission outside the Pakistan Central Corridor and Afghanistan North Corridor, detectable through ES but not AFP. Both methods revealed that viral diversity in early 2021 dropped to very low levels, with only a few lineages present, supporting the three clusters detected in 2023.

### Phylogeographic assessment of viral movement
Movement of poliovirus across the 16 geographic regions over the past decade was inferred using a discrete trait analysis (Methods) and showed repeated cyclical movement of poliovirus between the southern regions of both countries. This particularly affected the South Corridor regions and Karachi (Fig. 2, purple and dark green regions). The virus was also inferred to circulate and expand in the south linked to importation from the North Corridor regions (Fig. 2, deep blue regions). Interestingly, cyclical movement of the B lineage between Karachi and the South Corridor regions during the 2019–2020 outbreak was observed with the full dataset (Fig. 2, Supplementary video file 1), but not picked when restricting the analysis to AFP data alone (Supplementary Fig. S3). Indeed, almost all AFP detections in the South Corridor Afghanistan region (72/73) and the Karachi region (6/7) in 2019 and 2020 were linked to the now extinct A lineage.

Overall, 895 (95% HPD 867–922) transitions were inferred between regions over the full time-period when including all data (Fig. 3), compared to only 268 (95% HPD 253–283) without the inclusion of ES data (Supplementary Fig. S3). Inferring the start and end region for each transition suggests the city of Karachi has been the primary region seeding transmission in other regions over the decade of the study (Fig. 3B). This association is found both with and without the inclusion of ES data, although a significantly greater number of exportations are estimated when both AFP and ES data are considered (all data: 240 exportations; 95% HPD: 212–266, AFP data: 63 exportations; 95% HPD 40–82, Fig. 3B, Supplementary Fig. S3C). This analysis also highlights major historical importers of poliovirus, with the North and South Corridors of Pakistan being consistently high importers (Fig. 3A). Sindh and the East Pakistan region are found to be major importers only when ES data are included (Fig. 3A and Supplementary Fig. S3B), suggesting that many of these importations, which are only detected through ES, do not circulate for a long enough time to result in paralysis cases.

### Local transmission lineages
We define local transmission lineages (LTLs) as clusters of phylogenetically linked detections in the same geographic region (Fig. 4A shows descriptive diagrams of the splitting process and classification into distinct categories). Using these LTLs, the estimated mean time to detection of a virus imported into a region was 93 days (CI 85–101) overall across the whole period. Looking at the regions supporting

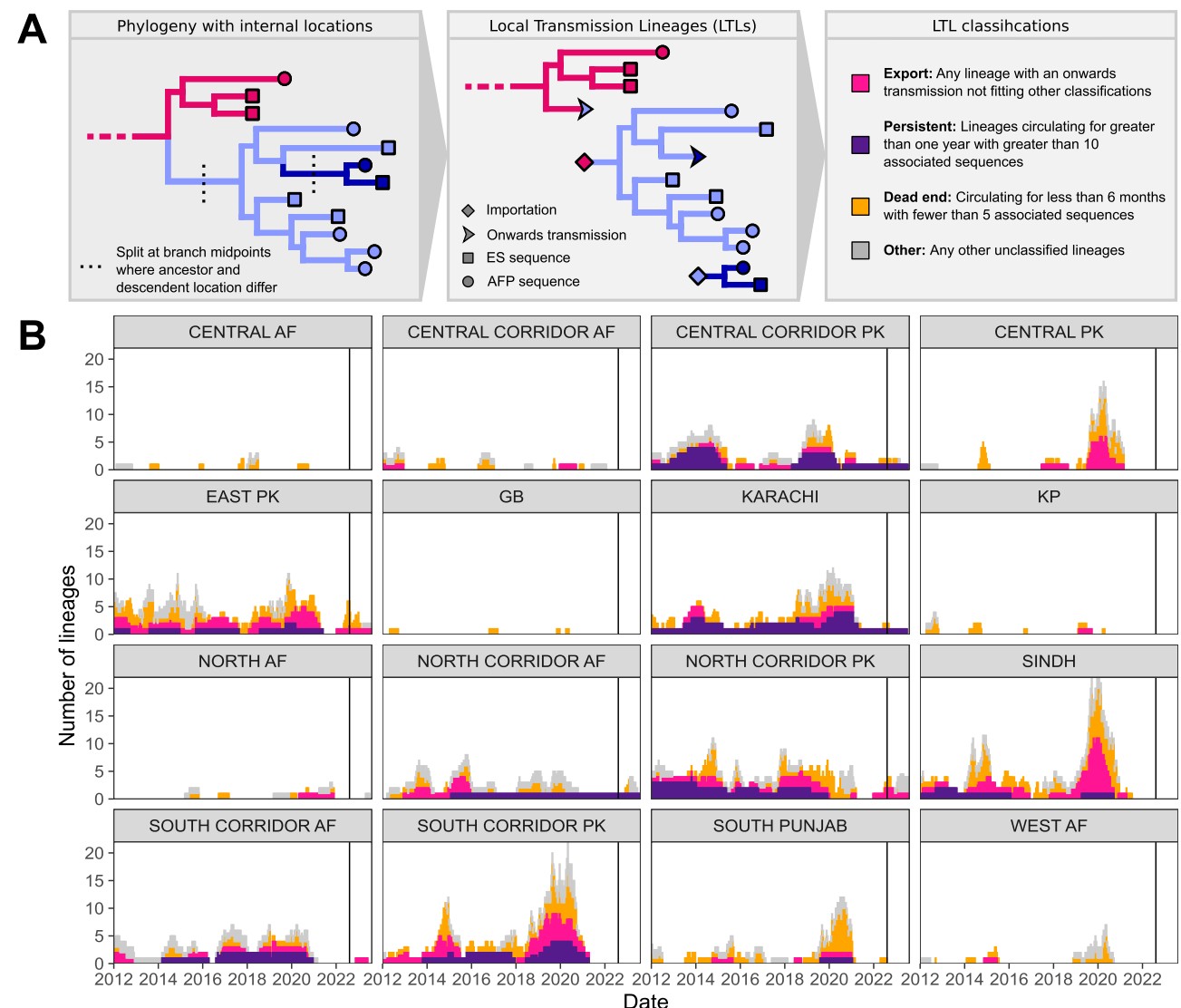

**Fig. 4 | Analysis of local transmission lineages. A** Flowchart showing the process of generating and classifying local transmission lineages (LTLs). **B** Classification of LTLs by region over the time period covered by the data. These plots show median numbers of lineages in each group over time over a posterior sample of 300 phylogenies. 95% CrIs for these estimates are shown in Supplementary Fig. S4. The black vertical lines are one year prior to the end of the dataset, after which we can't be sure of lineage classifications due to data censoring.

persistent transmission in 2023 (the Central Corridor Pakistan and North Corridor Afghanistan regions), this was 112 days (CI 89–134), compared to 92 days (CI 84–101) in the other regions. Without the early warning given by ES, we estimate an overall mean time to detection of an imported cluster of 166 days (CI: 144–188). This corresponds to an estimated average improvement of 73 days (equivalent to around 2 and a half months), in the time to detection of an emerging lineage with the ES system, over AFP surveillance alone.

We classified the reconstructed LTLs into four categories based on the number of detections, the duration of detection, and the number of exportation events associated with each lineage. These categories are 'Dead end' (circulating for less than 6 months with less than 5 associated sequences), 'Persistent' (circulating for greater than 1 year with greater than 10 associated sequences), 'Export' (Any lineage with onwards transmissions not falling into the previous categories and 'Other' (Any remaining lineages not classified). The median number of lineages falling into each category over time is shown in Fig. 4B, with the 95% HPD across 300 sampled phylogenies shown in the Supplementary Information (Supplementary Fig. S4). Median numbers

of lineages obtained with only AFP data are also provided in the Supplementary Information (Supplementary Fig. S5). Over the full period, an average of 72% (median: 588/816) of all detected LTLs across reconstructed phylogenies were dead ends, not leading to onwards transmission in other regions or persistent circulation, suggesting good population immunity across Pakistan and Afghanistan. Only the Central Corridor Pakistan (median: 8; CrI: 7–9), Karachi (median: 8, CrI: 6–11) and South Corridor Pakistan (median: 8, CrI: 6–10) regions were found to have significantly greater than 5 persistent LTLs, highlighting their historic importance as long term harbors of poliovirus transmission, and key regions to target for poliovirus eradication. Long-term persistence has also been supported in the two North Corridor regions on either side of the border, although with a smaller number of distinct persistent lineages.

Grouping the LTLs into categories also shows that persistent transmission in most regions was interrupted following the 2019–2020 outbreak (Fig. 4). Each of the North Corridor of Afghanistan and the Central Corridor of Pakistan regions, the current focal regions for polio eradication efforts in the region, are found to have harbored a single

persistent LTL each since early 2021. These regions supported the majority of the continued circulation of polio during the low transmission period (clusters 2 and 3 on Fig. 2A). Another persistent lineage was hosted in the city of Karachi, driven by a single detection in wastewater in mid-2023 (cluster 1 on Fig. 2A). This detection was highly divergent to other viruses in the region (5.4% divergent from any previously sequenced virus), with our analysis suggesting this isolate had a most recent common ancestor in May 2020, (2020-05-20, CrI 2020-01-25 to 2020-08-22), around 3 years before its detection. The reasons for this genetically distinct detection and any potential reservoir which supported its long-term persistence have yet to be identified. Similarly, although not included in this analysis, polioviruses distantly related to those circulating in southern Pakistan were detected in Mozambique and Malawi in 2022, highlighting the international risk posed by undetected polioviruses[3].

All recent circulation (apart from the single detection in Karachi in mid-2023) has been linked to the two distinct persistent chains of transmission in the North Corridor Afghanistan and Central Corridor Pakistan regions (clusters 2 and 3 on Fig. 2A), with spill-over events into other regions seen towards the end of the included data. Full plots showing which regions poliovirus was imported from and exported to for each LTL are provided in the Supplementary Information (Supplementary Fig. S6).

### Recurring patterns of spread

The 2013-14 and 2019-20 outbreaks occurred primarily in the same set of focal regions, with both outbreaks emerging from exportations from the North Corridor (Figs. 2A and 3C, D). This exportation is a pattern that repeats towards the end of the included data, in early 2023. This is less clear without ES data, where the main stem (the most common ancestral line) of the phylogenetic tree is inferred to move between regions more regularly (Supplementary Fig. S3A). Although large numbers of AFP cases are detected in the Central Corridor of Pakistan during both outbreaks, very little exportation is observed from this region into other areas (Fig. 3C, D). The role of the city of Karachi to amplify and repeatedly export poliovirus during outbreak periods, with cyclical movement between this region and the Southern Corridor on both sides of the border is also visible on the phylogenetic tree (Figs. 2A and 3C, D). This cyclical movement is again less clear without the inclusion of ES data, although we do infer exportations from both Karachi and the surrounding Sindh region into the South Corridor with AFP data alone.

### Sensitivity to sampling

To test the sensitivity of the full analysis to sampling differences between regions, we calculate adjusted Bayes factors for transitions between pairs of regions using a uniform randomized sample of locations at the tips. This is carried out using the tip-state-swap operator in BEAST[22,23]. This phylogeographic sensitivity analysis showed that the inferred root location of the whole tree and the most recent common ancestor of the A and B clades in the North Corridor of Pakistan is not explained by the sampling locations at the tips alone and is influenced by the genetic data. High sampling in the Karachi, North Corridor Pakistan and East Pakistan regions were found to likely be influencing the number of inferred transitions into and out of these regions. Overall, we see significantly fewer transitions between regions than expected under a null model, highlighting the importance of geographic clustering to the transmission of WPV1 in Pakistan and Afghanistan (Supplementary Table S2). A more complete description of this analysis and the findings is provided in the supplementary text.

Analyses using AFP data alone are considered here for comparison with the combined AFP and ES analysis, and the results using these data are presented in the Supplementary Information. Despite this, the non-Polio AFP reporting rate, which is independent of poliovirus transmission in each of the regions, increased over time in all regions and varied widely between regions (Supplementary Fig. S7). A method to incorporate sampling bias into estimates of movements between regions on the AFP tree was applied to polio data and presented in the Supplementary Information[24]. This method accounts for sampling rate differences between regions at the sampled tips, in a similar way to previous published work[25]. The results with this correction support the findings presented using the AFP data only, with all estimated rates falling within the bounds of the corrected analysis.

## Discussion

Routinely collected polio genetic sequence data has the potential to provide crucial insights to the Global Polio Eradication Initiative. Genetic data, compared to case data, contains information on the relatedness of viral isolates, which can be used to reconstruct viral movement[26]. In this study, we have unraveled historic patterns of polio spread and identified regions which act as importers and exporters of wild poliovirus in the last areas of endemic transmission over an 11-year period.

The approach used here was popularized by Dudas and colleagues in 2017, revealing the importance of human mobility and cross-border movement in the 2013-16 West African Ebola outbreak[19]. This approach is interesting for polio as previous analyses of polio genetic data have constructed clusters of related viruses without a geographic component[14,27,28]. Including geographic data provides insights into the direction of transmission, movement of lineages across regions, and patterns of spread, critical for informing public health interventions. Our results underscore the critical role of known reservoirs in the North and Central Corridor regions in maintaining poliovirus persistence between major outbreaks. The data suggest that sustained circulation within the North Corridor specifically, poses a significant risk of transmission expansion to other regions. These corridor regions are typically characterized by under-immunized populations, hard-to-access areas and challenges to healthcare delivery driven by insecurity, geography and socioeconomic factors; all of which contribute to their importance to polio transmission[7,29]. Following major vaccination efforts which brought the last major outbreak under control these regions were key to sustaining endemic polio circulation, as is clearly illustrated by the LTLs presented in this work[9]. This widespread vaccination was also likely a major factor in the extinction of the A lineage, which was only circulating in the more accessible southern regions at the time the outbreak was brought under control (Fig. 2).

The Karachi and South Corridor regions should also remain key targets for vaccination and surveillance, even when not supporting ongoing transmission, due to their role in amplifying and supporting the past two major outbreaks. Karachi, one of the world's most densely populated urban areas with poor sanitation in many parts of the city, supports large mobile and migrant populations, under-immunized communities and high risk areas for vaccination teams, making it a key melting pot for viral spread[30-32]. The importance of these regions is widely supported by previous spatial and genetic analyses of polio transmission in the region[11,29].

The analysis highlights the importance of the East Pakistan region, which includes five major population centers, as both an importer and exporter of poliovirus, often acting as a link between circulation in the north and south of Pakistan. This role is not seen when looking at AFP data alone, possibly suggesting that the burden is not high enough to result in significant numbers of paralytic polio cases in the region. This difference could also be due to the high coverage of routine immunization with inactivated poliovirus vaccine in this region, which protects infected individuals against paralysis, but does not prevent shedding of poliovirus in stool[33]. Detections in East Pakistan also underscore the utility of ES as a surveillance mechanism to reveal viral circulation even in the absence of clinical paralysis cases, which are a very rare outcome for polio-infected individuals[34]. Most detections in the East Pakistan region do not lead to sustained transmission, with

viral lineages in the region circulating for an average of 143 days (mean; CI: 114-172). It is possible that a large amount of shedding in wastewater is due to individuals in transit through these major cities, which would be of interest as an early marker of viral dispersion from endemic areas.

Comparing the full dataset to data only from AFP cases allows us to show the early warning predictive ability of ES for poliovirus transmission, as previously investigated by Cowger et al.[10]. Our study supports their finding of an improvement in the detection of circulation with the inclusion of wastewater surveillance sites for polio monitoring over AFP reporting alone, although to a lesser extent. Here we find a 2.5-month improvement in time to detection of imported poliovirus compared to a 4-month improvement in the previous study. This is likely driven by the differences in how transmission chains are classified between the two studies, with the previous study looking at the detection time of a genetically similar virus before and after each polio case with and without the inclusion of ES data, where here we generate an inferred importation time of virus into a region[9]. Using the date of the ancestral node as the date of importation, the most conservative estimate for time to detection in our model, a mixed ES and AFP sample provides a more comparable 3.7 month improvement over AFP sampling alone. The uncertainty of the timing of a movement between regions, and the likelihood of undetected movements along a branch scale with branch length, a limitation of all simple discrete trait reconstructions.

Incorporating ES data in a phylogenetic analysis for polio poses an issue for interpretation and lineage classification. Wastewater sites are generally in fixed locations and sampled at regular intervals, in contrast to stool samples from individuals with AFP, who are sampled directly from the human population. Differences in sampling intensity and targeting of ES sites to densely populated or high-risk areas need to be considered when interpreting the results of this analysis, as well as incomplete reporting of the viruses present in wastewater, particularly from sites with large and diverse catchment populations. Recent sampling in Pakistan has incorporated a larger number of 'adhoc' surveillance sites used for a short period of time in an attempt to track down key missed populations (These can be seen in recent years in Supplementary Fig. S2 which shows positive and negative samples by site by region). The data also highlights the fact that Pakistan has a much larger ES network than Afghanistan and may be considered more representative of underlying transmission, although the importance of capturing cross border movements of poliovirus precludes treating the ES systems in the two countries independently for analysis. The tip-state-swap phylogeographic analysis suggests the data are informative of movement patterns and past circulation of poliovirus but the oversampling associated with the intensity of ES surveillance in certain regions likely influences the patterns reconstructed. Sampling-independent phylogeographic methods such as the structured coalescent could be used to perform this type of analysis, although the number of regions which can be considered and computational time and power required impede their use for fast reporting, as required in an eradication context[35–37].

Incorporating data from wastewater samples is an increasingly important question for phylogenetics and phylodynamics, with wide-scale expansion in wastewater sampling following the global SARS-CoV-2 pandemic. The error estimation in this analysis is an improvement over current regularly reported poliovirus genetic analyses, but further work is needed to ensure robust results are provided to policymakers. Spatially biased data in simple continuous-time Markov chain methods such as the one reported here have previously been shown to lead to estimates with overly high confidence in the reported values, as well as decreasing accuracy with the degree to which data are biased[38]. A proposed adaptation to account for differential sampling over time is briefly presented in the Supplementary Information, allowing a correction for sampling differences in a simple phylogeographic model. Work to improve these methods and raise awareness of the assumptions made when treating location as a discrete trait is increasingly important as phylogeography becomes increasingly popular as an epidemiological tool[24,39,40].

Previous studies, including infectious disease epidemiological investigations, have looked at population movements within Pakistan with different sources of human mobility data, including mobile phone data and Meta/Facebook data for good movement indices[41–43]. These analyses broadly agree with the predominating patterns of movement in this work, linking Northeast to Southwest Pakistan[43]. These datasets introduce additional challenges for interpretation with their own biases, particularly tending to be less representative of rural areas which often support highly mobile populations. Molodecky et al. needed to supplement the available mobile phone data with a radiation model to be able to model these rural areas, which are key reservoirs of wild poliovirus persistence, as evidenced by repeated virus detections[42] Afghanistan has very little available movement data, with mostly low population density and large numbers of internally and internationally displaced individuals. Due to conflict, changing interaction between the two governments and other such metapopulation-level factors, human movement in this region over a decade would be difficult to accurately reconstruct and incorporate effectively in a simple phylogeographic model such as the one used here. The aim of this study is to highlight the additional information on the links between poliovirus detections which can be made using sequence data, links which cannot be made with case reporting alone. More complex phylodynamic models may be better suited to incorporating available additional data in future analyses, although these would likely need to be at a smaller spatial and temporal scale to allow for the incomplete spatial data available.

With the expansion of wild polio transmission in late 2023 and 2024, the number of cases has reached its highest level in five years. This resurgence poses significant challenges for polio eradication efforts in the last endemic countries. To achieve the goal of a polio-free world, sustained commitment, community engagement and healthcare security are essential. This analysis underscores that the regions with known accessibility issues, sanitation deficiencies and healthcare shortcomings are key to the persistence of wild poliovirus. Continued international cooperation is vital to overcoming barriers to vaccination and surveillance efforts in the region.

## Methods

In this analysis we infer phylogenies of wild serotype 1 poliovirus with genetic sequence data and use a continuous-time Markov Chain (CTMC) method to estimate the likely locations of internal nodes in these phylogenies[44]. To achieve this, we first reconstruct a set of empirical trees with BEAST 1.10.5 before implementing the Markov Jump method of Minin and Suchard to estimate ancestral locations and transitions between regions[22,44,45]. The R (version 4.4.1) pipeline created for this analysis recreates aspects of the baltic python pipeline (https://github.com/evogytis/baltic) with R code to generate migration maps, phylogenetic trees with overlaid case data, and local transmission lineages from a set of posterior phylogenies output from BEAST. The code is built on several R packages, primarily the phylogenetic package ape and the ggplot2 and ggtree plotting packages and is archived on github (https://github.com/JorgensenD/WPV1_genetics_2023)[46–48].

### Data

We analyzed VP1 gene sequences of serotype 1 wild poliovirus detected across Pakistan and Afghanistan through stool samples and ES surveillance. These sequences were generated by the Regional Reference Laboratory for Poliomyelitis at the National Institute of Health, Islamabad, Pakistan as part of routine poliovirus surveillance. This laboratory performs virus culture and sequencing for both Pakistan

and Afghanistan. The culture and sequencing of these viruses followed the WHO gold-standard protocol. Viruses which have a cytopathic effect are characterized by an intratypic differentiation reverse transcription quantitative PCR (ITD RT-qPCR) and poliovirus detections are referred for VP1 sequencing[17].

The sequences are 906 bases long with mostly synonymous amino acid changes which occur according to a molecular clock at a high rate compared with most viral pathogens[49–51]. The dataset used in this analysis includes 3,843 sequences which could be matched to location and date metadata and cover the period from 1st January 2012 to 28th August 2023 (Supplementary Table S1). Two sequences which could not be matched to metadata were removed prior to analysis. The two data sources, AFP and ES, have different collection methodologies, with numbers of AFP cases generally matching well with intensity of transmission. AFP data are considered a-priori less biased than ES sampling, which is restricted to specific sampling sites[20,34]. AFP data themselves can be biased due to differences in reporting rates by region, with some regions better served by healthcare services than others. We show non-polio AFP rates as a measure of surveillance intensity independent of polio transmission in Supplementary Fig. S7. ES samples can often return a large number of closely related or identical sequences due to the use of multiple cell culture flasks for each sample and the presence of viral mixtures, meaning there are duplicate sequences in the numbers included in Supplementary Table S1. Here we retain duplicate sequences as they do not affect inferred transitions although they could be removed to reduce computational cost. Where we estimate possible numbers of transitions in each outbreak, we exclude these zero length branches between pairs of identical sequences. The number of ES sites sampled has expanded over time, from 23 in 2012 (all in Pakistan) to 221 in 2022 (189 in Pakistan and 32 in Afghanistan) (Supplementary Figs. S1, S2). These sites are targeted to known reservoirs of WPV1 transmission, transportation hubs and areas with large mobile populations[52]. To assess the potential impact on the results of using sequences from ES, we performed phylogeographic analysis on two datasets, the full dataset including all sequences presented in the main text and an AFP-only dataset presented in the supplementary material.

## Outlying sequences
Prior to reconstruction of ancestral locations with discrete trait analysis, the sequence data were aligned with MAFFT (version 7.490) and checked for outliers[53]. A maximum likelihood phylogeny was generated with IQ-Tree (version 1.6.12) and re-rooted with the R package ape (version 5.3) to give a phylogeny most compatible with a strict molecular clock[46,54]. Root-to-tip divergence was calculated in R with EpiGenR (version 1.1.0) (Supplementary Fig. S8)[55].

## Discrete locations
In order to give a clear picture of geographic movement of poliovirus within the endemic countries, we divided them into 16 discrete geographic regions (Fig. 1B). These regions were informed by previous epidemiological analyses which highlighted the "corridor" areas between the two countries and the port city of Karachi in the south of Pakistan as key regions for poliovirus transmission and persistence[12,14,15,41,56]. This corridor designation is routinely used by the GPEI and partners. The corridors all straddle the border between the two countries, with our analysis further separating these regions by country; they are: the South Corridor, centered around the major border crossing between Quetta and Kandahar; the North Corridor, centered around a major border crossing between Peshawar and Jalalabad linking the capitals of the two countries; and the Central Corridor, incorporating a more porous border with various smaller crossing points. We further split Afghanistan according to previously used 'agro-climatic zones', merging northern regions which report few poliovirus cases (Fig. 1A)[57,58]. Outside of the corridor regions, Pakistan

is subdivided primarily by province, with North Punjab and surrounding regions in Azad Jammu and Kashmir and Islamabad grouped together and separated from the less densely populated south Punjab region[59].

## Bayesian phylogeny generation
Prior to Bayesian tree generation, a single maximum likelihood tree was inferred from the available sequence data with IQ-TREE[54]. From this initial tree, branches were collapsed and polytomies resolved randomly to give eight unique starting trees for Bayesian phylogeny generation. These starting trees were dated using the R package treedater (version 0.5.0) with the molecular clock rate estimated using initial bounds of 0.005 and 0.015 substitutions per site per year and a starting value of 0.01, informed by previous analyses of the VP1 region[27,60].

Eight Markov chains, initiated with each of the unique starting trees and a unique seed were run in parallel for 15 million iterations with a phylogeny sampled every 1000 iterations. These chains represent repeated analyses of the same set of genetic sequences with the aim of assessing the convergence of inferred values and parameters from different starting positions. The Bayesian phylogenies were generated under a simple coalescent constant size tree prior. A strict clock with an informative normal prior centered on a clock rate of 0.01 was used, again informed by previous analyses and the inferred root to tip regression. Multiple chains were used to speed computation and to allow convergence to be assessed by ensuring effective sample sizes for each parameter were over 200 and 95% credible intervals were not significantly different across chains. After the iteration limit was reached, the initial 30% of each chain was removed as burn-in and the tree files combined with the BEAST utility logcombiner[22]. For the complete dataset of ES and AFP data, two of the eight chains failed to converge and were dropped from further analysis. For the AFP dataset alone, all chains were retained. The trees produced were downsampled to give a set of 1000 trees to be used as an empirical tree set for DTA.

## Markov jumps and rewards (MJ)
Discrete trait transitions over the phylogeny were estimated using the Markov Jumps and Rewards (MJ) method of Minin and Suchard, included in BEAST 1.10.5[45]. This method was chosen over simpler discrete trait CTMC as it can be used to estimate numbers of transitions between locations directly in a BEAST model. Despite the large number of parameters to estimate, the method can be sped up significantly by supplying a pre-estimated empirical tree set to estimate over as we did in this analysis. As the numbers of transitions are sampled directly as parameters in BEAST, statistical support and bounds for numbers of transitions can be trivially calculated in post-analysis of the Markov Chain Monte Carlo (MCMC) trace files. MJ was run for 10 million iterations with one of the 1000 empirical trees randomly sampled at each iteration. A 10% burn-in was removed after the iteration limit was reached.

Spread3 v0.9.7 was used to assign Bayes Factor support to the inferred number of transitions between pairs of discrete locations[61]. Transitions with Bayes Factor support above 10 are presented in the results section.

## Measures of genetic diversity through time
We reconstructed viral diversity over time using two different approaches to account for the mixed sample of environmental surveillance (ES) and acute flaccid paralysis (AFP) data. For AFP data, which are sampled exclusively from the human population, we employed a Bayesian skyline model to estimate past population dynamics, reconstructing the effective population size over time $(Ne(t))$. This is produced using a stepwise-constant coalescent Bayesian skyline method on the available AFP sequences with 100 million

iterations and 50 segments in BEAST 1.10.5. Each segment contains the same number of coalescent events, meaning that segments represent shorter time periods when there is more branching activity occurring in the tree. The mean and 95% CrI over time are calculated from the posterior after removing a 10% burn-in in the BEAST utility Tracer. However, for the full dataset, which includes wastewater samples, the Bayesian skyline approach was not appropriate. Instead, we used a lineages through time (LTT) analysis, based on the number of branching events from the posterior of the empirical tree generation procedure outlined previously. The LTT plot was censored six months before the end of the data as we cannot differentiate between right censoring and lineage extinction.

### Consensus tree generation
To allow for the visualization of geographic clusters of transmission, the set of phylogenetic trees output from BEAST were combined to produce a representative consensus tree. A maximum clade credibility tree with common ancestor heights was generated with the BEAST utility TreeAnnotator, including location assignments at each internal node from MJ. Common ancestor heights were used to ensure only positive branch lengths were reconstructed in the final maximum clade credibility phylogeny. This consensus tree was used to produce the example representative phylogeny presented in the results section and example phylogenies presented in the supplementary Information.

### Local transmission lineages
When inferring local transmission lineages (LTLs) we use a sample of 300 phylogenies from the posterior of the MJ analysis. These phylogenies are split wherever a movement between regions is inferred (two connected nodes of the phylogeny are inferred to be in different regions). Here we use the midpoint of the branch between connected nodes with different inferred locations as the estimated time of this movement (Fig. 4). This technique was based on visualizations presented by Dudas et al. for the 2013-2016 West African Ebola outbreak[19].

LTLs across the sample of phylogenies are summarized by classifying them into three distinct types. The first, 'dead-end' lineages, are defined as circulating for less than 6 months with fewer than 5 associated sequences and without onwards transmission to other regions. The second, 'persistent' lineages are detected for greater than one year with greater than 10 associated sequences. The third are 'export' lineages which include any remaining lineages not included in the previous two groups which seed onwards transmission in another region. This is visualized in Fig. 4.

We calculate a mean time to detection and 95% CI for each region using the inferred LTLs. Time to detection is calculated as the time from the midpoint of the branch connecting an LTL to a node in a different region to the first detection in the LTL by any surveillance method. As there will be edge effects due to censoring of the data at the end of the period covered, with any extant lineages which are yet to be detected at the endpoint not included, we do not assign any lineages extant in the last 6-months of the data to the dead-end group.

### Inclusion and ethics statement
This work has been developed and produced in a collaboration between authors at the Gates Foundation, National Institutes of Health, Pakistan (NIH) and Imperial College London. The data remains the property of the NIH and WHO and is governed by their respective data sharing policies. As a secondary analysis of surveillance data we have gone to lengths to ensure anonymity is maintained throughout. Ethical review approval for secondary analysis of poliovirus surveillance data was obtained from the Imperial College Research Ethics Committee (Ref: 21IC6996).

### Reporting summary
Further information on research design is available in the Nature Portfolio Reporting Summary linked to this article.

## Data availability
Epidemiological data analyzed in this study were obtained from the World Health Organization (WHO) Polio Information System on 11th April 2024. These data are the property of the individual countries, and data access was provided through the Global Polio Eradication Initiative (GPEI) data sharing agreement. Data are available from the WHO Institutional Data Access/Ethics Committee for GPEI research partners who meet the criteria for access to confidential data. (https://extranet. who.int/polis/) The genetic sequence data, accessed on 10th September 2023, are provided by the National Institute of Health, Pakistan (NIH) for this collaborative study. These data are provided under a direct data sharing agreement between NIH Pakistan and Imperial College London. Requests for access to these data can be made directly to NIH at salman14m@gmail.com according to the NIH Pakistan's policy on data sharing using the reference number 1WPV12012308PKAF. Minimal working example data are provided to reproduce the figures included in the manuscript on GitHub https:// github.com/JorgensenD/WPV1_genetics_2023). This does not include any identifiable information or genetic sequence data. This repository is archived with Zenodo[62].

## Code availability
Code used for the analysis and visualisation of the data included in this manuscript is available on GitHub (https://github.com/JorgensenD/ WPV1_genetics_2023). This repository is archived with Zenodo[62].

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

## Acknowledgements

We acknowledge members of the Global Polio Laboratory Network (GPLN) and Global Polio Eradication Initiative (GPEI) who contribute to the collection and generation of poliovirus sequence and case data. We would also like to thank Dr. Isobel Blake (Imperial College London) and Dr. Natalia Molodecky for providing guidance on poliovirus surveillance and Dr. Fabiana Gámbaro (Université Libre de Bruxelles) for providing advice on the tip swapping sensitivity analysis for discrete trait analysis.D.J., M.P.S. and N.C.G. acknowledge funding from the MRC Centre for Global Infectious Disease Analysis (reference MR/X020258/1), funded by the UK Medical Research Council (MRC). This UK funded award is carried out in the frame of the Global Health EDCTP3 Joint Undertaking. D.J., M.P.S. and N.C.G. also acknowledge funding by Community Jameel. D.J. and M.P.S. are funded by the Wellcome Trust and Royal Society (216427/Z/19/Z) and N.C.G. is funded by the WHO Polio Research Committee. D.J. acknowledges PhD funding from UKRI EPSRC. M.P.S. is a Sir Henry Dale Fellow, a program jointly funded by the Wellcome Trust and the Royal Society (216427/Z/19/Z). H.L. and S.K. are employed by the Gates Foundation. M.M.A., M.S., A.K., Y.A. and N.M. are employed at the polio reference laboratory at the National Institutes for Health, Islamabad, Pakistan which is part funded by the World Health Organization (WHO).

## Author contributions

Conceptualization: D.J., M.P.S., N.C.G. Methodology: D.J., M.P.S., N.C.G., D.D.S.C. Investigation: D.J., M.P.S., N.C.G., M.M.A., M.S., A.K., Y.A., N.M., D.D.S.C. Data curation: M.M.A., M.S. Visualization: D.J. Supervision: M.P.S., N.C.G., M.M.A. Writing – original draft: D.J., M.P.S., N.C.G. Writing – review and editing: D.J., M.P.S., N.C.G., M.M.A., H.L., S.K., D.D.S.C.

## Competing interests

The authors declare no competing interests.
