## [Transparent Peer Review file · Nature Communications]

Evolution and Transmission Dynamics of Wild Poliovirus in Pakistan and Afghanistan (2012-2023)

Corresponding Author: Dr David Jorgensen

Version 0:

Reviewer comments:

Reviewer #1

(Remarks to the Author)

Overarching Comment: The authors present a novel phylogeographic analysis that show how polioviruses were exported, imported and sustained transmission across Pakistan and Afghanistan.

Line 43-44: phylogenetics is a type of surveillance so this sentence is saying surveillance informs surveillance, I would revise.

Line 81-86 ignore that Afghanistan has had bans on house to house campaigns which have greatly hindered vaccination efforts. This needs to be noted if discussing the reasons for unvaccinated children and transmission continuing.

123-128: Appreciate the sensitivity analyses, but the authors should consider looking at Afghanistan and Pakistan separately in terms of ES. The ES network, especially in the last few year in Pakistan is gigantic, and likely more "representative" than Afghanistan's network.

173-174: The authors should also consider the IPV coverage in these areas as well. High use of IPV will further suppress the paralysis to infection ratio making AFP cases much less likely while subclinical infections occur.

216-221: The orphan detected in Karachi serves as an important reminder that despite the extensive surveillance network in Pakistan, we manage to miss circulation. Please also note the WPV1 importation into Malawi/Mozambique was also an orphan.

249-251: I do not think its appropriate to say that AFP reporting is independent of transmission. During outbreaks or areas of concerns of endemic transmission surveillance efforts are intensified resulting in an increased sensitivity (and a lot of non-AFP reporting).

319-321: The sensitivity analyses should consider the differences between adhoc ES sites vs. established ES sites.

Pakistan started sampling using Ad hoc sites extensively in 2023 where collections occurred only 1-2 times compared to established sites. Would be good to note how that mixed approach and the oversampling of viruses effects this analysis.

341-343: Agree with the authors on the limitations but I have even greater concerns. Pakistan has managed to have orphans despite the vast ES network. Inherent in poliovirus phylogenetics is that you only have a sample of all viruses circulating. Therefore, assumptions made about the phylogeography need to take into account that you never can see everything. Policy makers are not always savvy consumers of this information so we need to make sure to not overstate the findings.

375-378: What proportion of sequences could not be matched to the geographic metadata?

466: Given the right censoring in the ES data, the analysis may not have included as many adhoc sites that proliferated in 2023. That said, it would be interesting to understand what proportion of the ES is from adhoc sites vs. well established sites.

Overarching Comment in relation to figure 4 and text in support of fig: We need a stronger explanation of the "other" category. The existing categories do not read as mutually exclusive and collectively exhaustive. Would other include lineages that circulated for >6m but <12m (i.e. not dead-ends and not persistent). If so, please explain as such. Additionally, please include a foot note or some explanation of how data are right censored in the figure (i.e. if an LTE was first identified in January 2023) it would not necessarily have "time" to be observed as a persistent transmission lineage.

Overarching Comment: Please speak to the branch length and the limitations in measuring "where" the virus was when the genetic distances between viruses is greater (not just orphans). This limitation, while uncommon in Pakistan and Afghanistan occurs and clouds our understanding of where the virus was geographically during the period of unobserved

evolution.

(Remarks on code availability)

Very well documented code with a clear ReadMe file. The authors correctly noted that reviewers will not be able to run all code without access to the POLIS database. As I was in the field without access to my regular computer with R, so I was unable to code myself.

Reviewer #2

(Remarks to the Author)

In the manuscript entitled «Evolution and Transmission Dynamics of Wild Poliovirus in Pakistan and Afghanistan (2012-2023): implications for global polio eradication», the authors performed a phylogeographic analysis using genetic sequence data in order to provide a comprehensive account of the recent history of wild poliovirus 1 (WPV1) circulation to an unprecedented temporal and spatial extent. The authors make use of genetic sequences collected in stool samples from paralytic poliovirus cases as well as from environmental surveillance in wastewater. The authors highlight the following main results of public health relevance:

- 1) Increase in viral diversity was associated with two major outbreaks, and the difference between lineages through time plots (using both wastewater and stool samples) and effective population size (using solely stool samples) strongly suggests that environmental surveillance could be used as an early-warning system.
- 2) Identifying in the phylogenetic tree parent and children nodes in different geographic units allowed estimating the date of movement between those units. As such, the combination of stool and wastewater samples leads to an improved estimation of time to detection of a virus imported into a geographic unit by 73 days.
- 3) Local transmission lineage analysis resulted in classification of phylogenetically linked clusters to be persistent, dead ends or exporting. Therefore, the authors could show that Karachi had persistent transmission during 2013-2023, with cyclical movement to adjacent regions, and that the two major outbreak foci were located in the North Corridor region.

Although genomic surveillance and phylogenetic analysis for WPV1 in Pakistan has been undertaken previously (e.g. Alam et al. CID 2016, Akhtar et al. CID 2019), the explicit use of phylogeographic approaches is a noteworthy novelty in the field. Persistent within-region circulation, sources and sinks of transmission have been identified and are highly relevant for monitoring and evaluation of the polio program. The present work also corroborates previous result suggesting the use of environmental surveillance as an early warning system for WPV1 transmission.

The methods are presented in sufficient detail, and the published code allows to reproduce the present analysis or carry out similar analyses.

I have several general comments and questions.

1) Validation of derived between-region transmission patterns: The chord diagrams in Fig 3 C&D but also Fig S7 are suggestive that the authors have identified viral movement patterns. It would be worthwhile to validate these patterns of virus movement with human mobility data (or models), as the authors seem to suggest in their manuscript that human movement could be the main driver of the observed transmission dynamics. Recent examples from SARS-CoV2 phylogeographic analysis (du Plessis et al. Science 2021, Lee et al. Emerg. Microb. Infection 2023) performing glm analysis for discrete trait diffusion could help reveal the drivers of observed viral movement patterns (e.g. population density, socio-economic factors, sanitation, human movement, immunization).

2) The choice of the discrete spatial traits and sampling bias of ES: The authors highlight the correction of sampling bias of AFP cases between different regions. From Table S1, it appears that there is also a striking imbalance between AFP and ES cases per region, owing presumably to high variability in the number of ES sites. This will certainly introduce a strong bias in the discrete trait analysis to correctly infer location and thus the movement between ES samples. The population fraction within ES catchment areas relative to the total population for each region could indicate whether regions are evenly covered in terms of ES. As far as I understand, ES sampling bias has not been corrected for.

The choice of the regions for the discrete trait analysis seems to be made such that it is operationally relevant to the polio program. How sensitive are the inferred viral movements towards the particular choice of regional boundaries, given the sampling biases between regions?

3) Population immunity: Global estimates (Voorman et al. Vaccine 2023) suggest that both Afghanistan and Pakistan were able to maintain exceptionally high mucosal immunity levels of above 90% against WPV1 from 2016 to 2020 across all regions. Do the authors assume that these levels were maintained until 2023? It would be helpful have some understanding on the data limitations for WPV1 immunity, and why immunity levels have not been used as discrete traits, as under-immunization has been mentioned in the manuscript for the corridor regions and Karachi.

(Remarks on code availability)

The code allows to reproduce the main figures of the manuscript. From the README file, it is clear how to proceed with the analysis.

Reviewer #3

(Remarks to the Author)

The study of Jorgensen et al. is particularly timely given the recent expansion of wild polio transmission in late 2023 and 2024, which has resulted in the highest number of cases in five years. This resurgence poses significant challenges to polio eradication. In this context, a historical perspective and an analysis of virus movement over time and space are crucial to guide interventions. The manuscript provides valuable insights into the direction of transmission, the movement of lineages across regions, and the patterns of wild poliovirus circulation in the last areas of endemic transmission (Pakistan and Afghanistan) over an 11-year period.

The study highlights the importance of incorporating a geographical component when analyzing polio genetic data. Additionally, the study underscores the value of incorporating wastewater sample data for phylogenetic and phylodynamic analyses. Adding environmental data, the authors observed a 2.5-month reduction in the time required to detect imported poliovirus. Additionally, the manuscript underscores the critical role of known reservoirs in the North and Central Corridor regions in maintaining poliovirus persistence between major outbreaks in 2013-2014 and 2019-2020. The authors also identify the Karachi and South Corridor regions as key targets for vaccination and surveillance. Although these regions may not support ongoing transmission, they play a significant role in amplifying and sustaining major outbreaks, as evidenced by their involvement in the past epidemics. The conclusions are supported by strong evidence and are essential for guiding polio eradication efforts in these two remaining endemic countries.

Overall, the manuscript is of significant interest, though some minor points require attention.

1. Line 63-64 in the "introduction": "Between 2012 and 2023, there were two major outbreaks of WPV1 in Pakistan and Afghanistan, around 2013-14 and 2019-2020 (Fig 1A)". This sentence would better correspond to the results.
2. The link provided in reference 4 is not accessible
3. Lines 69-72: "The persistence of wild poliovirus in Pakistan and Afghanistan, alongside vaccine-derived cases in Africa and Asia, has resulted in polio being designated a PHEIC". Regarding this phrase, in the WHO statement referenced in 2014, it was emphasized that the primary concern was not the presence of vaccine-derived polio cases, which were not specifically mentioned at that time. The key issue was the persistent transmission of wild poliovirus in endemic countries (particularly Pakistan, Afghanistan, and also Nigeria) which posed a significant risk to global eradication efforts. An additional reason for the PHEIC was that wild poliovirus was being exported internationally (from Pakistan to Afghanistan, from Syria to Iraq, and from Cameroon to Equatorial Guinea) leading to the reintroduction of polio in areas previously free from the disease. Therefore, I would suggest modifying the sentence explaining why polio was considered a PHEIC in 2014 to reflect the true reasons outlined by the WHO at the time.
4. Line 99 "acute flaccid paralysis (AFP)". The acronym AFP should be defined the first time it appears in the text (line 93)
5. Line 117: The figure numbers in the supplementary materials should be based on the order in which they appear in the text. For example, the first supplementary figure to appear is Fig. S6, which should actually be the first in the supplementary data section.
6. Line 119: "see map in Figure 1C". Shouldn't it be 1B?
7. Line 134-135 "with no significant difference when using AFP data alone". Does this refer to figure S3?
8. Line 164: What do the authors mean by the phrase "Splitting numbers of inferred transitions...".
9. Line 232: Should the colors inside the "circles" representing the links between regions in Figures 3C and 3D correspond to those in Figures 3A and 3B (or in other words, to those in the legend)? When looking at Figures 3C and 3D, I assume the light and dark blue represent North Corridor AF and PK, respectively, but they do not match the legend.
10. Line 142 in Supp: Where it says "alternative representation of the data plotted in figure 2 of the main text", shouldn't it say "alternative representation of the data plotted in figure 3 of the main text"?

(Remarks on code availability)

Reviewer #4

(Remarks to the Author)

In their manuscript, Jorgensen and colleagues explored the recent history of wild poliovirus 1 (WP1) spread across Afghanistan and Pakistan over the last decade. They analyzed genetic sequences of WP1 isolated in acute flaccid paralysis cases (AFP) or in wastewater over 2014-2023 using Bayesian phylogeographic models to identify hubs of transmission. They obtained similar results when using AFP cases only or AFP+wastewater samples and showed that the corridor regions and Karachi play a major role in WP1 persistence and diffusion. The authors also highlighted that lineages may persist over several years without detection which stresses the need for long-term global surveillance efforts. With their results the authors hope to provide an evidence-based picture of the transmission patterns of WP1 over the last decade to improve surveillance efforts and guide vaccination campaigns. Overall, the manuscript is well-written. Sensitivity analyses were carried out to investigate whether and how the results are

impacted by inherent biases of the surveillance system and the hypotheses underlying the phylogeographic models used. The discussion is expanded, and the authors remain cautious in the interpretation of their results. Although I recognize the scientific and public health relevance of the study as well as the quality of the work, some aspects of the methods remain unclear, and more emphasis should be made (i) on how the results were used to guide polio vaccination and surveillance and (ii) on the perspectives of this work.

Please find my detailed comments below.

1. In their abstract, the authors state that “This information [the results of the study] has been used to guide polio vaccination and surveillance”. However, the authors do not detail in the rest of the manuscript how the surveillance system was improved and where and when vaccination campaigns were rolled out based on their results. Besides, if the authors had their first results that included the wastewater samples before the start of the 2024 outbreak, did they rely on their results to guide the response in real-time? By adding such information, the authors will make a stronger case of the use of phylogeographic tools to provide early warning signals.
2. In the introduction, it is stated that “overall reported coverage of vaccination campaigns remains high”. Could the authors provide estimates of the vaccine coverage or a reference?
3. Wastewater surveillance provided more than half of the samples included in the main analysis. However, very few details were given on this surveillance system. Could the authors provide more information such sampling frequency by site, a bar plot of the number of sites monitored per year, whether the wastewater sites included hospitals or not, average time between sampling and sequencing...? A similar question applies to the surveillance of AFP cases (are all detected cases sequenced, average time between sampling and sequencing).
4. In the methods, it is unclear which tree generating process was used to analyze the AFP+ environmental samples data. On a similar note, did the authors test the Bayesian skygrid instead of the Bayesian skyline model? Did it change the estimated viral population dynamics?
5. The authors report a strong correlation between the number of sequences and the number of AFP cases by region across regions and years. Which test was used?
6. In the results, the authors report cyclical movements between the Southern corridor and Karachi which is very difficult to see on the figures. A video that could be made available on the github repository would better illustrate this statement.
7. In the supplementary materials, the authors mention that AFP surveillance probably vary across regions using non-polio AFP reporting rates as an example. As a non-specialist of polio, I am wondering whether we could expect differences in the proportion of severe (AFP) cases among all polio cases across regions due to differences in underlying population health levels?
8. Polio being a strictly human disease, large-scale spread of the disease is primarily mediated by human movement. Did the authors have access to mobility data? Would it be difficult to have access to such data? Did the authors investigate which covariates are associated with between-region diffusion rates using a generalized linear model of diffusion (original paper PMID: PMC3930559)?
9. Another interesting aspect was little discussed: what are the hypotheses of the authors concerning the extinction of lineage A? Could it be related to different transmission rates across lineages? Tools such as BDSKY λ could help evaluate these differences (original paper PMID: PMC10725310).

(Remarks on code availability)

Would it be possible to make the genetic sequences, as well as their metadata, on the github repository? This would ensure transparency of the main and sensitivity analyses.

Besides the codes of the sensitivity analysis presented in the Supplementary Text where the authors correct for the biased sampling of the AFP cases is not available. Could it be added to the repository?

Version 1:

Reviewer comments:

Reviewer #2

(Remarks to the Author)

I thank the authors for addressing my comments and improving the manuscript especially by including the sensitivity analysis.

(Remarks on code availability)

I have not installed or run the code, but I checked the availability of code, e.g. for the tip swapping analysis. The README file seems to me sufficient to reproduce the analysis.

Reviewer #3

(Remarks to the Author)

Points raised have been adequately addressed in the revised version of the manuscript

(Remarks on code availability)

Reviewer #4

(Remarks to the Author)

I thank the authors for their detailed responses. I am satisfied with most of the changes made to the manuscript, supplementary materials, and data and code sharing statements based on my concerns.

I have three final comments:

1. The authors report using Pearson correlation to evaluate the correlation between the number of sequences and the number of cases across years and across regions. Unless I'm mistaken, the Pearson correlation is calculated for two continuous variables in a specific region. Then, there should be 19 coefficients for the years, and 16 for the regions. In my opinion, the revised sentence is misleading. An alternative would be to report in the main text that the correlation is high and ranges between X and X for years and X and X for regions, then add a figure or table with the details. Besides, a non-parametric correlation analysis using Spearman correlation or Kendall's tau correlation coefficients is more appropriate to evaluate the correlation for count data (unless they are normally distributed, or sample sizes are large enough).
2. The authors have added a tip-state-swap analysis to identify the transition links that come up significant in the discrete phylogeographic analysis probably due to the geographical sampling bias. It should be written as "tip-state-swap discrete phylogeographic analysis" instead of "tip swap" or "tipswap analysis" (see results and methods of the main manuscript, and supplementary materials).
3. Minor comment on line 387: I would not mention continuous phylogeography because, in my opinion, it is not appropriate to model human population migration, whether at the national or regional scale. Indeed, human mobility is mostly constrained by roads and driven by gradients in population density. Models used in continuous phylogeography are more appropriate when animal or plant hosts play a role in the epidemiological cycle of the pathogen.

Otherwise, I would be happy to see the manuscript published in Nature Communications.

(Remarks on code availability)

The authors made the relevant changes in the code availability statement.

REVIEWER COMMENTS

Reviewer #1 (Remarks to the Author):

Overarching Comment: The authors present a novel phylogeographic analysis that show how polioviruses were exported, imported and sustained transmission across Pakistan and Afghanistan.

Line 43-44: phylogenetics is a type of surveillance so this sentence is saying surveillance informs surveillance, I would revise.

In this context we agree that phylogenetic analysis is a surveillance tool within the broader surveillance network, which was the intended meaning of this sentence. We have revised the text to read:

L 37-40:

“Phylogenetic analysis has long been central to the polio surveillance network, and advancing the approaches used can provide critical epidemiological insights to accelerate eradication efforts.”

Line 81-86 ignore that Afghanistan has had bans on house to house campaigns which have greatly hindered vaccination efforts. This needs to be noted if discussing the reasons for unvaccinated children and transmission continuing.

This is a good additional point. We have added this to the previous sentence on insecurity:

L70-73:

“Insecurity in the region, including armed conflict, intermittent bans on house-to-house vaccinations, and targeted attacks on vaccinators, have made certain regions and communities difficult or impossible to access for health workers, allowing polio transmission to persist despite overall increases in vaccination coverage.”

123-128: Appreciate the sensitivity analyses, but the authors should consider looking at Afghanistan and Pakistan separately in terms of ES. The ES network, especially in the last few year in Pakistan is gigantic, and likely more “representative” than Afghanistan’s network.

We agree with the reviewer that the ES network in Pakistan has expanded at a much faster rate in recent years, especially with the introduction of ad-hoc sampling sites. A key part of the continued transmission of poliovirus in the region is thought to be linked to the border regions and the interplay between populations on either side of the border, leading us to consider the two countries as one interconnected population. The regions used do not

bridge the border between the two countries to give a better representation of cross-border movement.

The following clarifying sentence has been added to the manuscript, reiterating the importance of considering the two countries together when investigating viral movement patterns:

L345-348:

“The data also highlights the fact that Pakistan has a much larger ES network than Afghanistan and may be considered more representative of underlying transmission, although the importance of capturing cross border movements of poliovirus precludes treating the ES systems in the two countries independently for analysis.”

To further investigate the differences in ES between the two countries, we have added a supplementary figure showing the number of ES samples through time by site and region (Fig. S2). Numbers of sampled ES sites through time in each region remain fairly consistent up to the end of the last major outbreak and the introduction of large numbers of ‘ad-hoc’ ES sites in Pakistan. None of these new sites report positive detections in the time-period covered in this analysis.

An additional sensitivity analysis has been added to the revised manuscript to incorporate sampling rate differences across regions using a tip swapping approach. This approach performs a uniform exchange of the locations at the tips of the phylogeny, maintaining the proportion sampled from each location but switching which tips are sampled from where. This allows us to build a null model of the sampling bias in the data, without the additional information drawn from the sequence data itself. This null model can then be used to correct the original analysis by adjusting the Bayes Factors for transitions between pairs of locations. It also allows us to compare the inferred locations at ancestral nodes in the tree to see if these inferences are due solely to sampling biases or informed by the genetic data. The following text has been added to the supplement, as well as a supplementary table comparing inferred movements under each model (Table S2):

Supplementary text, paragraph 2+3:

“Tipswap analysis

The tipswap model uses the ‘TipStateSwapOperator’ operator in BEAST to uniformly resample the locations at the tips of the phylogeny at each iteration to give a null prior distribution of movements between the locations. This null model is informed by the geographic sample used in the analysis but not the genetic data. The null model places the root of the full phylogeny in Sindh (pp = 0.156) and the root of all recent detections) when excluding early samples which appear to fall into a separate ancestral clade) in

the south corridor region of Pakistan (pp =0.164). Both ancestors are found to be most probably in the north corridor region of Pakistan (pp = 0.255 and 0.615 respectively) in our analysis. This difference suggests that the locations of these ancestors is informed by the genetic data rather than biased sampling by location. We see the same difference for the ancestor of the clade B (tipswap ancestor: North Corridor PK, pp = 0.192, inferred ancestor North Corridor AF, pp = 0.989) labelled in figure 2. For clade A, the null and inferred ancestral locations are both in Karachi, although the posterior probability is much higher in our analysis (tipswap pp = 0.316, inferred pp = 0.883).

The null model provided by the tipswap analysis can also be used to correct the Bayes factors for the number of movements between regions inferred in the analysis. By correcting the Bayes factors using the posterior probabilities of each transition pathway under the tipswap and inference models, we can account for transitions where significance is heavily influenced by regionally biased sampling. With the corrected Bayes factors under the tipswapping approach, 43 pairs of regions are found to have strongly significant ($BF \geq 10$) transitions informing the model. This is down from 66 found to be strongly significant without correcting the Bayes factors. Most significantly, nine transitions out of Karachi to other regions and six into Karachi are found to be explainable under the null model informed by the regional sampling rates as are five out of the North Corridor region of Pakistan and two into this region. These two regions have the most samples in the analysis, with 16.7% and 15.4% of the total samples coming from Karachi and the North Corridor of Pakistan respectively.”

The results of this sensitivity analysis are also included in the main text:

L244-255:

“To test the sensitivity of the full analysis to sampling differences between regions, we calculate adjusted Bayes factors for transitions between pairs of regions using a uniform randomized sample of locations at the tips. This is carried out using the tip state swap operator in BEAST.¹ This sensitivity analysis showed that the inferred root location of the whole tree and the most recent common ancestor of the A and B clades in the North Corridor of Pakistan is not explained by the sampling locations at the tips alone and is influenced by the genetic data. High sampling in the Karachi, North Corridor Pakistan and East Pakistan regions were found to likely be influencing the number of inferred transitions into and out of these regions. Overall, we see significantly fewer transitions between regions than expected under a null model, highlighting the importance of geographic clustering to the transmission of WPV1 in Pakistan and Afghanistan (Table S2). A more complete description of this analysis and the findings is provided in the supplementary text.”

And briefly discussed alongside the AFP-only analysis:

L348-351:

“The tipswap sensitivity analysis suggests the data are informative of movement patterns and past circulation of poliovirus but the oversampling associated with the intensity of ES surveillance in certain regions likely influences the patterns reconstructed.”

173-174: The authors should also consider the IPV coverage in these areas as well. High use of IPV will further suppress the paralysis to infection ratio making AFP cases much less likely while subclinical infections occur.

This is an insightful alternate interpretation of the ES-specific detections in Punjab. The research group at Imperial College London plans to analyse and map the coverage of polio vaccine types in Pakistan in a separate piece of work which would be useful to inform this type of analysis in the future. A published estimate from Molodecky et al., et al. Vaccine 2023 (<https://doi.org/10.1016/j.vaccine.2021.09.037>) shows clear differences in routine vaccination coverage across the region. In Pakistan, the routine schedule is made up of both oral and inactivated polio vaccines and unfortunately we don't currently have access to IPV dose reporting figures. We have added this as a further point in the article:

L308-311:

“This difference could also be due to the high coverage of routine immunization with inactivated poliovirus vaccine in this region, which protects infected individuals against paralysis, but does not prevent shedding of poliovirus in stool. ²”

216-221: The orphan detected in Karachi serves as an important reminder that despite the extensive surveillance network in Pakistan, we manage to miss circulation. Please also note the WPV1 importation into Malawi/Mozambique was also an orphan.

We have added a further sentence highlighting these detections. However, these detections are not included in the submitted work due to data ownership restrictions.

L217-219:

“Similarly, although not included in this analysis, polioviruses distantly related to those circulating in southern Pakistan were detected in Mozambique and Malawi in 2022, highlighting the international risk posed by undetected polioviruses.”

249-251: I do not think its appropriate to say that AFP reporting is independent of transmission. During outbreaks or areas of concerns of endemic transmission surveillance efforts are intensified resulting in an increased sensitivity (and a lot of non-AFP reporting).

Due to the severe nature of AFP cases, we expect them to be more robust to sampling changes than virus detections through ES. We agree that the description of this assumption in the original manuscript is lacking. We have updated the text to remove the term “proxy unbiased”. Additional analysis correcting for AFP detection rates is also presented, which supports the findings in the main analysis.

The sentence in the manuscript has been changed to:

L257-259:

“Analyses using AFP data alone are considered here for comparison with the combined AFP and ES analysis, and the results using these data are presented in the Supplementary Information.”

319-321: The sensitivity analyses should consider the differences between adhoc ES sites vs. established ES sites. Pakistan started sampling using Ad hoc sites extensively in 2023 where collections occurred only 1-2 times compared to established sites. Would be good to note how that mixed approach and the oversampling of viruses affect this analysis.

Ad-hoc surveillance sites are an important change to the sampling framework. To assess the impact on this analysis we have extracted additional data from the WHO polio information system (POLIS) on reported ES samples. Although samples are reported from ad-hoc sites during the time-period covered, none are found to be positive for poliovirus. We have included a figure showing the sites and samples per region (Fig. S2) and have added the following text discussing this change in sampling to the manuscript:

L341-344:

“Recent sampling in Pakistan has incorporated a larger number of ‘ad-hoc’ surveillance sites used for a short period of time in an attempt to track down key missed populations (These can be seen in recent years in Figure S2 which shows positive and negative samples by site by region.)”

341-343: Agree with the authors on the limitations but I have even greater concerns. Pakistan has managed to have orphans despite the vast ES network. Inherent in poliovirus phylogenetics is that you only have a sample of all viruses circulating. Therefore, assumptions made about the phylogeography need to take into account that you never can see everything. Policy makers are not always savvy consumers of this information so we need to make sure to not overstate the findings.

We think the limitations and some possible corrections are described clearly in the text. We aim, primarily, to describe the additional information provided by sequence data over simple case data, which also represent a sample of symptomatic cases. The existence of

orphan and long branch detections, only reconstructed with sequence data, demonstrate gaps in surveillance more clearly than any other poliovirus surveillance data.

375-378: What proportion of sequences could not be matched to the geographic metadata?

Two sequences were removed as they could not be matched to metadata (<0.01%). This has been added to the text:

L576:

“Two sequences which could not be matched to metadata were removed prior to analysis.”

466: Given the right censoring in the ES data, the analysis may not have included as many adhoc sites that proliferated in 2023. That said, it would be interesting to understand what proportion of the ES is from adhoc sites vs. well established sites.

We point the reviewer to the figures generated in response to the previous mention of ad-hoc sites (Figure S2). This is an interesting point and may become increasingly important in future analyses of polio surveillance data.

Overarching Comment in relation to figure 4 and text in support of fig: We need a stronger explanation of the “other” category. The existing categories do not read as mutually exclusive and collectively exhaustive. Would other include lineages that circulated for >6m but <12m (i.e. not dead-ends and not persistent). If so, please explain as such. Additionally, please include a foot note or some explanation of how data are right censored in the figure (i.e. if an LTE was first identified in January 2023) it would not necessarily have “time” to be observed as a persistent transmission lineage.

The other category includes any LTLs which fall into this intermediate circulating time period or do not meet the required numbers of detections AND do not lead to onwards transmissions to other regions. “Other” is not intended as a coherent grouping but rather an NA classification for those lineages not falling into the described classifications. Therefore, the categories are mutually exclusive and exhaustive.

We have updated figure 4 to show a 1-year period where we cannot be certain of the classification of “other” lineages which may become persistent. We also cannot be certain that dead-end detections are truly dead-ends within the final 6 months of the dataset so these are reclassified as other. New versions of the two plots showing these classifications (Fig. 4, Fig. S5) have been added to the manuscript.

Overarching Comment: Please speak to the branch length and the limitations in measuring “where” the virus was when the genetic distances between viruses is greater (not just orphans). This limitation, while uncommon in Pakistan and Afghanistan occurs and clouds our understanding of where the virus was geographically during the period of unobserved evolution.

As we assign only a single location change at the midpoint of a branch we agree that the certainty of the timing of changepoint, and the likelihood of additional location changes not considered here, varies with the length of the branches. As a Bayesian analysis is used this reconstruction, with multiple phylogenies and discrete dispersion reconstructions considered, long branches explicitly lead to greater uncertainty in the results presented and less weighting towards the tips of these branches in the analysis. Despite the higher uncertainty in the timing or number of transitions which occur on a long branch, we can still infer that the two locations are linked, wether directly or indirectly.

We have added the following discussion point to highlight this:

L331-333:

“The uncertainty of the timing of a movement between regions, and the likelihood of undetected movements along a branch scale with branch length, a limitation of all simple discrete trait reconstructions.”

Reviewer #1 (Remarks on code availability):

Very well documented code with a clear ReadMe file. The authors correctly noted that reviewers will not be able to run all code without access to the POLIS database. As I was in the field without access to my regular computer with R, so I was unable to code myself.

Reviewer #2 (Remarks to the Author):

In the manuscript entitled «Evolution and Transmission Dynamics of Wild Poliovirus in Pakistan and Afghanistan (2012-2023): implications for global polio eradication», the authors performed a phylogeographic analysis using genetic sequence data in order to provide a comprehensive account of the recent history of wild poliovirus 1 (WPV1) circulation to an unprecedented temporal and spatial extent. The authors make use of genetic sequences collected in stool samples from paralytic poliovirus cases as well as from

environmental surveillance in wastewater. The authors highlight the following main results of public health relevance:

1) Increase in viral diversity was associated with two major outbreaks, and the difference between lineages through time plots (using both wastewater and stool samples) and effective population size (using solely stool samples) strongly suggests that environmental surveillance could be used as an early-warning system.

2) Identifying in the phylogenetic tree parent and children nodes in different geographic units allowed estimating the date of movement between those units. As such, the combination of stool and wastewater samples leads to an improved estimation of time to detection of a virus imported into a geographic unit by 73 days.

3) Local transmission lineage analysis resulted in classification of phylogenetically linked clusters to be persistent, dead ends or exporting. Therefore, the authors could show that Karachi had persistent transmission during 2013-2023, with cyclical movement to adjacent regions, and that the two major outbreak foci were located in the North Corridor region.

Although genomic surveillance and phylogenetic analysis for WPV1 in Pakistan has been undertaken previously (e.g. Alam et al. CID 2016, Akhtar et al. CID 2019), the explicit use of phylogeographic approaches is a noteworthy novelty in the field. Persistent within-region circulation, sources and sinks of transmission have been identified and are highly relevant for monitoring and evaluation of the polio program. The present work also corroborates previous result suggesting the use of environmental surveillance as an early warning system for WPV1 transmission.

The methods are presented in sufficient detail, and the published code allows to reproduce the present analysis or carry out similar analyses.

Thank you for these positive remarks.

I have several general comments and questions.

1) Validation of derived between-region transmission patterns: The chord diagrams in Fig 3 C&D but also Fig S7 are suggestive that the authors have identified viral movement patterns. It would be worthwhile to validate these patterns of virus movement with human mobility data (or models), as the authors seem to suggest in their manuscript that human movement could be the main driver of the observed transmission dynamics. Recent examples from SARS-CoV2 phylogeographic analysis (du Plessis et al. Science 2021, Lee et al. Emerg. Microb. Infection 2023) performing glm analysis for discrete trait diffusion could

help reveal the drivers of observed viral movement patterns (e.g. population density, socio-economic factors, sanitation, human movement, immunization).

We thank the reviewer for raising this point and agree that, ideally, human mobility data would be incorporated in our work or used for validation. However, these data are limited and often low quality for the region of interest when studying wild poliovirus transmission. We have added additional points to the discussion and raised possible future work with more complex models which could attempt to account for covariates driving the patterns of virus movement seen, particularly when analyses are carried out on smaller scales of time and space where more complete movement data may be available.

We have added the following paragraph on spatial data to the discussion section of the manuscript:

L370-389:

“Previous studies, including infectious disease epidemiological investigations, have looked at population movements within Pakistan with different sources of human mobility data, including mobile phone data and Meta/Facebook data for good movement indices.³⁻⁵ These analyses broadly agree with the predominating patterns of movement in this work, linking Northeast to Southwest Pakistan⁵. These datasets introduce additional challenges for interpretation with their own biases, particularly tending to be less representative of rural areas, which often support highly mobile populations. Molodecky et al. needed to supplement the available mobile phone data with a radiation model³ to be able to model rural areas which³ are key reservoirs of wild poliovirus persistence, as evidenced by repeated virus detections. Afghanistan has very little available movement data, with mostly low population density and large numbers of internally and internationally displaced individuals. Due to conflict, changing interaction between the two governments and other such metapopulation-level factors, human movement in this region over a decade would be difficult to accurately reconstruct and incorporate effectively in a simple phylogeographic model such as the one used here. The aim of this study is to highlight the additional information on the links between poliovirus detections which can be made using sequence data, links which cannot be made with case reporting alone. More complex phylodynamic models or continuous phylogeographic models may be better suited to incorporating available additional data in future analyses, although these would likely need to be at a smaller spatial and temporal scale to allow for the incomplete spatial data available.”

2) The choice of the discrete spatial traits and sampling bias of ES: The authors highlight the correction of sampling bias of AFP cases between different regions. From Table S1, it

appears that there is also a striking imbalance between AFP and ES cases per region, owing presumably to high variability in the number of ES sites. This will certainly introduce a strong bias in the discrete trait analysis to correctly infer location and thus the movement between ES samples. The population fraction within ES catchment areas relative to the total population for each region could indicate whether regions are evenly covered in terms of ES. As far as I understand, ES sampling bias has not been corrected for.

The choice of the regions for the discrete trait analysis seems to be made such that it is operationally relevant to the polio program. How sensitive are the inferred viral movements towards the particular choice of regional boundaries, given the sampling biases between regions?

As the reviewer states, there is no correction performed for ES sampling in the current analysis. The catchment populations of individual ES sites are difficult to calculate, particularly in populations without well-defined sewerage networks. A calculation of this sort would require additional assumptions to be made about both the catchment of a particular ES site and the population density within the catchment. The main findings of the work based only on the AFP data - without the ES data - as presented in the supplementary information - are consistent with the results obtained when ES data are included, although with less granularity. As stated in this comment, these more granular conclusions are subject to biases in the ES network.

In the revised manuscript we include an additional sensitivity analysis using a tip swapping approach. This approach performs a uniform exchange of the locations at the tips of the phylogeny, maintaining the proportion sampled from each location but switching which tips are sampled from where. This allows us to build a null model of the sampling bias in the data, without the additional information drawn from the sequence data itself. This null model can then be used to correct the original analysis by adjusting the Bayes Factors for transitions between pairs of locations. It also allows us to compare the inferred locations at ancestral nodes in the tree to see if these inferences are due solely to sampling biases or informed by the genetic data. The following text has been added to the supplement, as well as a supplementary table comparing inferred movements under each model (Table S2):

Supplementary text, paragraph 2+3:

“Tipswap analysis

The tipswap model uses the ‘TipStateSwapOperator’ operator in BEAST to uniformly resample the locations at the tips of the phylogeny at each iteration to give a null prior distribution of movements between the locations. This null model is informed by the

geographic sample used in the analysis but not the genetic data. The null model places the root of the full phylogeny in Sindh (pp = 0.156) and the root of all recent detections) when excluding early samples which appear to fall into a separate ancestral clade) in the south corridor region of Pakistan (pp = 0.164). Both ancestors are found to be most probably in the north corridor region of Pakistan (pp = 0.255 and 0.615 respectively) in our analysis. This difference suggests that the locations of these ancestors is informed by the genetic data rather than biased sampling by location. We see the same difference for the ancestor of the clade B (tipswap ancestor: North Corridor PK, pp = 0.192, inferred ancestor North Corridor AF, pp = 0.989) labelled in figure 2. For clade A the null and inferred ancestral locations are both in Karachi, although the posterior probability is much higher in our analysis (tipswap pp = 0.316, inferred pp = 0.883).

The null model provided by the tipswap analysis can also be used to correct the Bayes factors for the numbers of movements between regions inferred in the analysis. By correcting the Bayes factors using the posterior probabilities of each transition pathway under the tipswap and inference models, we can account for transitions where significance is heavily influenced by regionally biased sampling. With the corrected Bayes factors under the tipswapping approach 43 pairs of regions are found to have strongly significant ($BF \geq 10$) transitions informing the model. This is down from 66 found to be strongly significant without correcting the Bayes factors. Most significantly, nine transitions out of Karachi to other regions and six into Karachi are found to be explainable under the null model informed by the regional sampling rates as are five out of the North Corridor region of Pakistan and two into this region. These two regions have the most samples in the analysis, with 16.7% and 15.4% of the total samples coming from Karachi and the North Corridor of Pakistan respectively.”

The results of this sensitivity analysis are also included in the main text:

L244-255:

“To test the sensitivity of the full analysis to sampling differences between regions, we calculate adjusted Bayes factors for transitions between pairs of regions using a uniform randomized sample of locations at the tips. This is carried out using the tip state swap operator in BEAST.⁶ This sensitivity analysis showed that the inferred root location of the whole tree and the most recent common ancestor of the A and B clades in the North Corridor of Pakistan is not explained by the sampling locations at the tips alone and is influenced by the genetic data. High sampling in the Karachi, North Corridor Pakistan and East Pakistan regions were found to likely be influencing the number of inferred transitions into and out of these regions. Overall, we see significantly fewer transitions between regions than expected under a null model, highlighting the importance of geographic clustering to the transmission of WPV1 in Pakistan and Afghanistan (Table

S2). A more complete description of this analysis and the findings is provided in the supplementary text.”

And briefly discussed alongside the AFP-only analysis:

L348-351:

“The tipswap sensitivity analysis suggests the data are informative of movement patterns and past circulation of poliovirus but the oversampling associated with the intensity of ES surveillance in certain regions likely influences the patterns reconstructed.”

Discrete regions are an inherent assumption of the method used and regions were selected due to their importance in poliovirus epidemiology, with movements between them which are informative for the eradication initiative. These regions were discussed with stakeholders both regionally and internationally and are based on those used for routine analyses within the eradication initiative. Certain regions such as Central Pakistan, GB, KP West Afghanistan and North Afghanistan are specifically selected due to the small amount of environmental surveillance and small number of detected cases meaning it would be inappropriate to group these together with more sampled and populous regions.

3) Population immunity: Global estimates (Voorman et al. Vaccine 2023) suggest that both Afghanistan and Pakistan were able to maintain exceptionally high mucosal immunity levels of above 90% against WPV1 from 2016 to 2020 across all regions. Do the authors assume that these levels were maintained until 2023? It would be helpful have some understanding on the data limitations for WPV1 immunity, and why immunity levels have not been used as discrete traits, as under-immunization has been mentioned in the manuscript for the corridor regions and Karachi.

WPV1 immunity levels at the regional level, as presented by Voorman et al. are estimated to stay close to 100% due to the high frequency of immunisation activities carried out in the region. As both the immunisation activities and the administrative regions cover large areas, certain populations can be repeatedly missed, resulting in immunity gaps. There can be heterogeneous coverage within the regions if there are populations within them which are, for example: hard to access, not accepting of vaccinations, or highly mobile. These chronically missed populations would not be seen in global estimates of population immunity but repeated long chains of undetected transmission (seen as long branches in the phylogeny), as well as continued circulation despite large scale vaccination, suggest they must exist.

Reviewer #2 (Remarks on code availability):

The code allows to reproduce the main figures of the manuscript. From the README file, it is clear how to proceed with the analysis.

Reviewer #3 (Remarks to the Author):

The study of Jorgensen et al. is particularly timely given the recent expansion of wild polio transmission in late 2023 and 2024, which has resulted in the highest number of cases in five years. This resurgence poses significant challenges to polio eradication. In this context, a historical perspective and an analysis of virus movement over time and space are crucial to guide interventions. The manuscript provides valuable insights into the direction of transmission, the movement of lineages across regions, and the patterns of wild poliovirus circulation in the last areas of endemic transmission (Pakistan and Afghanistan) over an 11-year period.

The study highlights the importance of incorporating a geographical component when analyzing polio genetic data. Additionally, the study underscores the value of incorporating wastewater sample data for phylogenetic and phylodynamic analyses. Adding environmental data, the authors observed a 2.5-month reduction in the time required to detect imported poliovirus. Additionally, the manuscript underscores the critical role of known reservoirs in the North and Central Corridor regions in maintaining poliovirus persistence between major outbreaks in 2013-2014 and 2019-2020. The authors also identify the Karachi and South Corridor regions as key targets for vaccination and surveillance. Although these regions may not support ongoing transmission, they play a significant role in amplifying and sustaining major outbreaks, as evidenced by their involvement in the past epidemics. The conclusions are supported by strong evidence and are essential for guiding polio eradication efforts in these two remaining endemic countries.

Overall, the manuscript is of significant interest, though some minor points require attention.

Thank you for these positive comments.

1. Line 63-64 in the "introduction": "Between 2012 and 2023, there were two major outbreaks of WPV1 in Pakistan and Afghanistan, around 2013-14 and 2019-2020 (Fig 1A)".

This sentence would better correspond to the results.

We have made the suggested edit.

2. The link provided in reference 4 is not accessible

We thank the reviewer for pointing out this stale link. This link has now been updated.

3. Lines 69-72: "The persistence of wild poliovirus in Pakistan and Afghanistan, alongside vaccine-derived cases in Africa and Asia, has resulted in polio being designated a PHEIC". Regarding this phrase, in the WHO statement referenced in 2014, it was emphasized that the primary concern was not the presence of vaccine-derived polio cases, which were not specifically mentioned at that time. The key issue was the persistent transmission of wild poliovirus in endemic countries (particularly Pakistan, Afghanistan, and also Nigeria) which posed a significant risk to global eradication efforts. An additional reason for the PHEIC was that wild poliovirus was being exported internationally (from Pakistan to Afghanistan, from Syria to Iraq, and from Cameroon to Equatorial Guinea) leading to the reintroduction of polio in areas previously free from the disease. Therefore, I would suggest modifying the sentence explaining why polio was considered a PHEIC in 2014 to reflect the true reasons outlined by the WHO at the time.

This sentence has been updated to reflect the initial statement.

L63-66:

"The persistence of wild poliovirus and associated risk of international spread led polio to be designated a Public Health Emergency of International Concern (PHEIC) by the World Health Organization in 2014."

4. Line 99 "acute flaccid paralysis (AFP)". The acronym AFP should be defined the first time it appears in the text (line 93)

This has now been corrected in the text.

5. Line 117: The figure numbers in the supplementary materials should be based on the order in which they appear in the text. For example, the first supplementary figure to appear is Fig. S6, which should actually be the first in the supplementary data section.

This has been fixed and amended to incorporate the additional supplementary figures and tables.

6. Line 119: "see map in Figure 1C". Shouldn't it be 1B?

This is correct and has been updated in the text.

7. Line 134-135 “with no significant difference when using AFP data alone”. Does this refer to figure S3?

These values are not directly linked to the consensus trees shown in the figures but rather the full set of reconstructed phylogenies so we have not referenced the figures for either of the values presented.

8. Line 164: What do the authors mean by the phrase “Splitting numbers of inferred transitions...”?

This has been updated in the text to make the meaning clearer:

L159-160:

“Inferring the start and end region for each transition...”

9. Line 232: Should the colors inside the “circles” representing the links between regions in Figures 3C and 3D correspond to those in Figures 3A and 3B (or in other words, to those in the legend)? When looking at Figures 3C and 3D, I assume the light and dark blue represent North Corridor AF and PK, respectively, but they do not match the legend.

The colours within the circles are the same colours with a transparency applied to allow the crossing lines to be more easily discerned. The legend has been updated to show both the fully saturated and transparent colour (Fig. 3)

10. Line 142 in Supp: Where it says “alternative representation of the data plotted in figure 2 of the main text”, shouldn't it say “alternative representation of the data plotted in figure 3 of the main text”?

We thank the reviewer for pointing out this error, it has now been corrected.

Reviewer #4 (Remarks to the Author):

In their manuscript, Jorgensen and colleagues explored the recent history of wild poliovirus 1 (WP1) spread across Afghanistan and Pakistan over the last decade. They analyzed genetic sequences of WP1 isolated in acute flaccid paralysis cases (AFP) or in wastewater over 2014-2023 using Bayesian phylogeographic models to identify hubs of transmission. They obtained similar results when using AFP cases only or AFP+wastewater samples and showed that the corridor regions and Karachi play a major role in WP1 persistence and diffusion. The authors also highlighted that lineages may persist over several years without detection which stresses the need for long-term global surveillance efforts. With their results the authors hope to provide an evidence-based picture of the transmission patterns of WP1

over the last decade to improve surveillance efforts and guide vaccination campaigns.

Overall, the manuscript is well-written. Sensitivity analyses were carried out to investigate whether and how the results are impacted by inherent biases of the surveillance system and the hypotheses underlying the phylogeographic models used. The discussion is expanded, and the authors remain cautious in the interpretation of their results. Although I recognize the scientific and public health relevance of the study as well as the quality of the work, some aspects of the methods remain unclear, and more emphasis should be made (i) on how the results were used to guide polio vaccination and surveillance and (ii) on the perspectives of this work.

Please find my detailed comments below.

1. In their abstract, the authors state that "This information [the results of the study] has been used to guide polio vaccination and surveillance". However, the authors do not detail in the rest of the manuscript how the surveillance system was improved and where and when vaccination campaigns were rolled out based on their results. Besides, if the authors had their first results that included the wastewater samples before the start of the 2024 outbreak, did they rely on their results to guide the response in real-time? By adding such information, the authors will make a stronger case of the use of phylogeographic tools to provide early warning signals.

To date, these results have only been presented in annual meetings (The Technical Advisory Group on Polio Eradication for Afghanistan and Pakistan) which are used to plan the regional focus for the year ahead. Although this does inform strategy, we have removed the mentioned point from the abstract as it was also raised by another reviewer that surveillance (this work) guiding surveillance was not a well-articulated sentence.

We believe there is space for real-time analysis using methods such as this. Simple neighbour-joining phylogenetic analyses are routinely used within the polio eradication initiative when assigning genetic clusters, a key monitoring tool. Numbers of nucleotide changes between pairs of sequences are also used to assess the time over which a virus may have been circulating undetected and to find so called "orphan lineages". We believe the addition of routine phylogeographic analysis to capture virus movement and highlight any uncertainty is of critical importance to the eradication programme.

2. In the introduction, it is stated that "overall reported coverage of vaccination campaigns remains high". Could the authors provide estimates of the vaccine coverage or a reference?

We have added a reference to the work of Voorman et al. 2023 which includes population immunity estimates. Further information on individual vaccination campaign coverage is available from the WHO polio information system (POLIS), available to Global Polio Eradication Initiative research partners which we have also referenced.^{7,8}

3. Wastewater surveillance provided more than half of the samples included in the main analysis. However, very few details were given on this surveillance system. Could the authors provide more information such sampling frequency by site, a bar plot of the number of sites monitored per year, whether the wastewater sites included hospitals or not, average time between sampling and sequencing...? A similar question applies to the surveillance of AFP cases (are all detected cases sequenced, average time between sampling and sequencing).

For ES, plots of detections by region and site are now provided in the supplementary material (Figure S2). This shows the number of sites in each region, how often they are sampled and how many positive detections they have. We do not have accurate information on the catchment areas of the included ES sites, although they are usually designed to capture large areas with easy to access sites such as drainage channels. Some sites cover regions with healthcare facilities although not directly targeted to these facilities.⁹All AFP cases found to be positive for wild-type poliovirus by culture are sequenced. Sequencing is carried out according to gold-standard WHO protocols, taking around 2-3 weeks from sampling to sequencing.

4. In the methods, it is unclear which tree generating process was used to analyze the AFP+ environmental samples data. On a similar note, did the authors test the Bayesian skygrid instead of the Bayesian skyline model? Did it change the estimated viral population dynamics?

The fixed phylogenies are generated under a coalescent constant size tree prior. The tree generation process does not include the discrete trait for location. Intuitively, we would use a Bayesian skyline model due to the outbreaks described, but as we analyse data from both environmental and clinical surveillance, this method is not appropriate. We have found that the strongly clock-like evolution of the VP1 region of poliovirus leads to very similar tree reconstructions using a wide variety of methods. Minor differences in tree structure are unlikely to significantly alter the inference of viral movement from the included sequences, the main aim of this analysis. We have added the tree prior used to the methods section: L474-475:

“The Bayesian phylogenies were generated under a simple coalescent constant size tree prior”

We use the Bayesian skyline model when only including paralysis case data and this appears to accurately reconstruct known outbreaks of poliovirus in the region. This model is provided as a simple visual assessment of viral diversity. Although the skygrid method could provide a detailed reconstruction, this was not the main aim of this work.

5. The authors report a strong correlation between the number of sequences and the number of AFP cases by region across regions and years. Which test was used?

A Pearson correlation test was used. We have updated the text to reflect this:

L119

“Nevertheless, Pearson’s Correlation analysis revealed a strong correlation between the number of sequences and the number of reported WPV1 AFP cases across regions ($R = 0.75$, $p = 0.007$) and across years ($R = 0.82$, $p = 0.002$).”

This association was measured across regions and years independently, not by region across regions and years.

6. In the results, the authors report cyclical movements between the Southern corridor and Karachi which is very difficult to see on the figures. A video that could be made available on the github repository would better illustrate this statement.

We thank the reviewer for this suggestion. An animated map has been added to the github page and referenced as a supplementary file.

7. In the supplementary materials, the authors mention that AFP surveillance probably vary across regions using non-polio AFP reporting rates as an example. As a non-specialist of polio, I am wondering whether we could expect differences in the proportion of severe (AFP) cases among all polio cases across regions due to differences in underlying population health levels?

It is hard to assess these relationships in the case of polio where the case to infection ratio is very low and we don’t fully understand the risk factors for developing AFP once infected.

The most likely factor in the differential development of severe infections is the coverage of inactivated poliovirus vaccine (IPV), used in routine childhood vaccinations. IPV prevents severe outcomes for vaccinated individuals but does not prevent infection and onwards transmission.

The research group at Imperial College London plans to analyse and map the coverage of polio vaccine types in Pakistan in a separate piece of work which would be useful to inform this type of analysis in the future. A published estimate from Molodecky et al., et al. Vaccine 2023 (<https://doi.org/10.1016/j.vaccine.2021.09.037>) shows clear differences in routine vaccination coverage across the region, which often uses IPV. Unfortunately, we don't currently have access to IPV dose reporting figures.

We have added this as a further point in the article when referring to ES-specific detection of poliovirus in Punjab, a region with high reported routine immunization coverage:

L308-311:

“This difference could also be due to the high coverage of routine immunization with inactivated poliovirus vaccine in this region, which protects infected individuals against paralysis, but does not prevent shedding of poliovirus in stool. ²”

8. Polio being a strictly human disease, large-scale spread of the disease is primarily mediated by human movement. Did the authors have access to mobility data? Would it be difficult to have access to such data? Did the authors investigate which covariates are associated with between-region diffusion rates using a generalized linear model of diffusion (original paper PMID: PMC3930559)?

Based on this comment and the comments from other reviewers we have added a paragraph on human mobility to the discussion section. We thank the reviewers for raising this point and agree that, ideally, human mobility data would be incorporated in our work or used for validation. However, these data are limited and often low quality for the region of interest for wild type polio transmission. We have added additional points to the discussion and raised possible future work with more complex models which could attempt to account for covariates driving the patterns of virus movement seen, particularly when analyses are carried out on smaller scales of time and space.

We have added the following paragraph on spatial data to the discussion section of the manuscript:

L370-389:

“Previous studies, including infectious disease epidemiological investigations, have looked at population movements within Pakistan with different sources of human mobility data, including mobile phone data and Meta/Facebook data for good movement indices.³⁻⁵ These analyses broadly agree with the predominating patterns of movement in this work, linking Northeast to Southwest Pakistan⁵. These datasets introduce additional challenges for interpretation with their own biases, particularly tending to be less representative of rural areas which often support highly mobile populations.

Molodecky et al. needed to supplement the available mobile phone data with a radiation model³ to be able to model these rural areas, which are key reservoirs of wild poliovirus persistence, as evidenced by repeated virus detections. Afghanistan has very little available movement data, with mostly low population density and large numbers of internally and internationally displaced individuals. Due to conflict, changing interaction between the two governments and other such metapopulation-level factors, human movement in this region over a decade would be difficult to accurately reconstruct and incorporate effectively in a simple phylogeographic model such as the one used here. The aim of this study is to highlight the additional information on the links between poliovirus detections which can be made using sequence data, links which cannot be made with case reporting alone. More complex phylodynamic models or continuous phylogeographic models may be better suited to incorporating available additional data in future analyses, although these would likely need to be at a smaller spatial and temporal scale to allow for the incomplete spatial data available.”

9. Another interesting aspect was little discussed: what are the hypotheses of the authors concerning the extinction of lineage A? Could it be related to different transmission rates across lineages? Tools such as BDSKY λ could help evaluate these differences (original paper PMID: PMC10725310).

The two lineages circulated primarily in two separate regions between 2012 and 2019, with lineage A primarily seen in the more accessible southern regions. During the 2019-2020 outbreak, lineage A did not take hold outside of the southern regions whereas lineage B was much more widespread. Response to this outbreak was incredibly intense with large numbers of vaccination campaigns and it appears that the geographic spread of the B lineage was key to its persistence, allowing it to remain in remote reservoirs during the almost complete die out of both lineages in early 2021. We have added the following text:

L291-293:

“This widespread vaccination was also likely a major factor in the extinction of the A lineage, which was only circulating in the more accessible southern regions at the time the outbreak was brought under control (Fig.2).”

Reviewer #4 (Remarks on code availability):

Would it be possible to make the genetic sequences, as well as their metadata, on the github repository? This would ensure transparency of the main and sensitivity analyses.

Unfortunately, due to the public health sensitivity of this data it is not possible to upload to any public repository. A new data access statement has been drafted outlining the access process.

Besides the codes of the sensitivity analysis presented in the Supplementary Text where the authors correct for the biased sampling of the AFP cases is not available. Could it be added to the repository?

This method is described in a preprint and can be accessed via a separate github repository. We have added links to this code in the manuscript and github page.

References:

1. Vogels, C. B. F. *et al.* Phylogeographic reconstruction of the emergence and spread of Powassan virus in the northeastern United States. *Proc Natl Acad Sci U S A* **120**, e2218012120 (2023).
2. Molodecky, N. A. *et al.* Modelling the spread of serotype-2 vaccine derived-poliovirus outbreak in Pakistan and Afghanistan to inform outbreak control strategies in the context of the COVID-19 pandemic. *Vaccine* **41**, A93–A104 (2023).
3. Molodecky, N. A. *et al.* Risk factors and short-term projections for serotype-1 poliomyelitis incidence in Pakistan: A spatiotemporal analysis. *PLoS Med* **14**, e1002323 (2017).
4. Rowe, F. Using digital footprint data to monitor human mobility and support rapid humanitarian responses. *Reg Stud Reg Sci* **9**, 665–668 (2022).
5. Wesolowski, A. *et al.* Impact of human mobility on the emergence of dengue epidemics in Pakistan. *Proc Natl Acad Sci U S A* **112**, 11887–11892 (2015).
6. Suchard, M. A. *et al.* Bayesian phylogenetic and phylodynamic data integration using BEAST 1.10. *Virus Evol* **4**, (2018).
7. Voorman, A. *et al.* Analysis of population immunity to poliovirus following cessation of trivalent oral polio vaccine. *Vaccine* **41 Suppl 1**, A85–A92 (2023).
8. World Health Organization. Polio Information System (POLIS). extranet.who.int/polis/public/CaseCount.aspx (2014).

9. Shaw, A. G. *et al.* Rapid and Sensitive Direct Detection and Identification of Poliovirus from Stool and Environmental Surveillance Samples by Use of Nanopore Sequencing. *J Clin Microbiol* **58**, e00920-20 (2020).

REVIEWERS' COMMENTS 2

Reviewer #2 (Remarks to the Author):

I thank the authors for addressing my comments and improving the manuscript especially by including the sensitivity analysis.

We thank reviewer 2 for raising this additional analysis and think the inclusion has improved the manuscript.

Reviewer #2 (Remarks on code availability):

I have not installed or run the code, but I checked the availability of code, e.g. for the tip swapping analysis. The README file seems to me sufficient to reproduce the analysis.

We have tried to make the README as comprehensive as possible alongside the beast templates to allow the work to be replicated.

Reviewer #3 (Remarks to the Author):

Points raised have been adequately addressed in the revised version of the manuscript

We thank the reviewer for the original comments.

Reviewer #4 (Remarks to the Author):

I thank the authors for their detailed responses. I am satisfied with most of the changes made to the manuscript, supplementary materials, and data and code sharing statements based on my concerns.

I have three final comments:

1. The authors report using Pearson correlation to evaluate the correlation between the number of sequences and the number of cases across years and across regions. Unless I'm mistaken, the Pearson correlation is calculated for two continuous variables in a specific region. Then, there should be 19 coefficients for the years, and 16 for the regions. In my opinion, the revised sentence is misleading. An alternative would be to report in the main text that the correlation is high and ranges between X and X for years and X and X for regions, then add a figure or table with the details. Besides, a non-parametric correlation analysis using Spearman correlation or Kendall's tau correlation coefficients is more appropriate to evaluate the correlation for count data (unless they are normally distributed, or sample sizes are large enough).

We have updated the correlation analysis to incorporate these comments. We selected Kendall's Tau to better incorporate the non-linearity of the relationship between the two measures. An additional supplementary table is included with the Kendall correlations across regions and years.

2. The authors have added a tip-state-swap analysis to identify the transition links that come up significant in the discrete phylogeographic analysis probably due to the geographical sampling bias. It should be written as "tip-state-swap discrete

phylogeographic analysis” instead of “tip swap” or “tipswap analysis” (see results and methods of the main manuscript, and supplementary materials).
This has been updated throughout.

3. Minor comment on line 387: I would not mention continuous phylogeography because, in my opinion, it is not appropriate to model human population migration, whether at the national or regional scale. Indeed, human mobility is mostly constrained by roads and driven by gradients in population density. Models used in continuous phylogeography are more appropriate when animal or plant hosts play a role in the epidemiological cycle of the pathogen.

This is an interesting discussion point but, as the reviewer suggests, not one we would like to raise in this analysis. We have removed the reference to continuous models.

Otherwise, I would be happy to see the manuscript published in Nature Communications.

We thank the reviewer for their work on this manuscript and providing additional comments which improve the interpretability of the manuscript.

Reviewer #4 (Remarks on code availability):

The authors made the relevant changes in the code availability statement.